# Instance-wise Feature Grouping

*Aria Masoomi[1], *Chieh Wu[1], Tingting Zhao[1], Zifeng Wang[1], Peter Castaldi[2], and Jennifer Dy[1]

*masoomi.a@northeastern.edu, wu.chie@northeastern.edu, t.zhao@northeastern.edu*
*zifengwang@ece.neu.edu, repjc@channing.harvard.edu, jdy@ece.neu.edu*
[1]*Department of Electrical and Computer Engineering, Northeastern University, Boston, MA, US*
[2]*Channing Division of Network Medicine, Brigham and Women's Hospital, Boston, MA, US*

## Abstract

In many learning problems, the domain scientist is often interested in discovering the groups of features that are redundant and are important for classification. Moreover, the features that belong to each group, and the important feature groups may vary per sample. But what do we mean by feature redundancy? In this paper, we formally define two types of redundancies using information theory: *Representation* and *Relevant redundancies*. We leverage these redundancies to design a formulation for instance-wise feature group discovery and reveal a theoretical guideline to help discover the appropriate number of groups. We approximate mutual information via a variational lower bound and learn the feature group and selector indicators with Gumbel-Softmax in optimizing our formulation. Experiments on synthetic data validate our theoretical claims. Experiments on MNIST, Fashion MNIST, and gene expression datasets show that our method discovers feature groups with high classification accuracies.

## 1 Introduction

Data samples are typically represented by features that domain experts assume to be important for a learning problem; however, not all features are important. The goal of feature selection is to select which features are needed to improve learning performance. Moreover, knowing which features are important helps in understanding learning algorithms.

Traditionally, *Feature Selection* algorithms find a *global* set of features for the entire data [1, 2, 3, 4, 5, 6, 7, 8, 9, 10]. While knowing the most important global features are useful, feature importance may vary across the entire population. For example in images, while one set of pixels may help us identify a shoe, a vastly different set of pixels would be required to identify a shirt. From this observation, there is an additional need for *Feature Selection* to be on a case-by-case basis, an approach also known as *Instance-wise Feature Selection*. A novel concept that has only been recently investigated in the context of explaining black-box models [11, 12, 13, 14]. Learning saliency maps [15] in some ways also provide some form of instance-wise feature importance by highlighting (weighting) important pixels in an image.

While *Instance-wise Feature Selection* focuses on each feature's relationship to its labels, it ignores the interaction among features. Multiple features may be equally important and yet redundant in relation to each other. Traditional feature selection algorithms (such as LASSO [16]) tend to select just one of these redundant features. However, in some domains such as gene expression applications, we are interested not only in which genes (features) are important but also in which genes interact together for disease prediction. Therefore, in addition to *Instance-wise Feature Selection*, we wish to also group the features based on their relationship with each other and to the label. There exist

methods like group Lasso (GLasso) [17] that selects which feature groups are important given a predefined grouping. Yet, in many applications the feature groups are unknown. Thus, methods that learn feature groups have been proposed [18, 19, 20, 21]. While these methods perform group feature selection, the groups are global and not instance-wise. In contrast to these approaches, this paper introduces *instance-wise* methods that can learn the feature group structure and identify its importance for prediction from an information theory perspective. We refer to this approach as *Instance-wise Feature Grouping*.

**Our Contribution.** We introduce a novel method for learning instance-wise feature grouping, the *group Interpreter (gI)*. Our formulation is made possible by our theoretical contribution of defining the concept of redundancy in this setting. Leveraging mutual information's ability to measure dependency, we formally define two types: *Representation Redundancy* captures the dependency between features while *Relevant Redundancy* captures the dependency between features and its corresponding labels. We prove how these redundancies can be captured and describe the mechanisms by which information is preserved. Our analysis leads to a lower bound to identify the number of groups for each sample. Moreover, we provide a practical algorithm that approximates mutual information (MI) through a variational lower bound. The algorithm also learns a mapping function that identifies the most important feature groups on a sample by sample basis. Finally, our theories are experimentally verified on both synthetic and real data from ongoing research. Indeed, our method reveals the difference in gene expression based on smoking status. We make the source code publicly available at `https://github.com/ariahimself/Instance-wise-Feature-Grouping`.

**Related Work.** Many traditional *global* feature selection utilizes MI as criterion for selection (as it is a natural criterion for measuring dependency among random variables). However, in global feature selection, the goal is to find the minimal subset of features relevant for prediction [4, 5, 6, 22, 1, 23]. A way to achieve finding this minimal subset is to maximize feature relevance while minimizing feature redundancy [24, 25]. Note that they wish to *remove* redundancy. In contrast, our goal is to learn which features group (i.e., cluster) together, where we define similarity of features based on *redundancy*. For example, mRMR [25] maximizes feature relevance while minimizing feature redundancy. If features F1 and F2 are highly dependent and relevant to prediction, only one will be chosen. In contrast, gI would select both as a group, *highlighting to domain scientists that these two features are both relevant and redundant to each other*. Unlike traditional feature clustering, our goal is to learn feature similarity not just based on their redundancy with each other (representation redundancy) but also on their redundancy based on their prediction ability (relevant redundacy). We formally define these concepts in this paper.

Among other global feature group learning methods, Chormunge and Jena [19] learn feature groups based on $k$-means clustering then apply gLasso; Bilevel Learning [20] learns the feature groups through a multi-task learning setting using bilevel optimization; OSCAR [18] automatically learns the feature groups by encouraging equality in the magnitude of each pair of variables. All these group feature selection methods are *global*; whereas, our proposed method gI learns feature groups *instance-wise*.

## 2 Ingredients of Feature Group Learning

**Overall Framework.** We summarize the overall network framework of our method in Figure 1, followed by a description of each component in this section.

Given data set $\mathbf{X} \in \mathbb{R}^{n \times d}$ with $n$ samples and $d$ features, and let $\mathbf{Y} \in \mathbb{R}^n$ be its corresponding labels; the $i^{\text{th}}$ sample input and its label are denoted as $x_i \in \mathbb{R}^d$ and $y_i \in \mathbb{R}$. Our goal is to separate the features into $k$ non-overlapping groups and select the $m$ most important groups for each sample. We learn for each sample a matrix $G$ to indicate each feature's group membership. The $G$ matrix is specifically constrained such that $G \in \{0, 1\}^{k \times d}$ where $G_{i,j} = 1$ if the $j^{\text{th}}$ feature of a sample belongs to the $i^{\text{th}}$ group. After compressing the features into $k$ groups, we also learn a vector $\mathbf{s} \in \{0, 1\}^k$ where $\mathbf{s}_\mu = 1$ if the $\mu$ group is among the $m$ most important groups.

**The Group Membership Matrix** $G$**.** We denote $\mathcal{G}$ as a random variable over a set of all possible $G$ matrices where $\mathcal{G} \in \{G \in \{0, 1\}^{k \times d} \mid \sum_{i=1}^k G_{ij} = 1\}$. This allows us to generate instance-wised $G$ matrices for each sample by learning $P(\mathcal{G}|\mathbf{X})$, and use its most likely outcome as $G$ where $G = \arg\max_{\mathcal{G}} P(\mathcal{G}|\mathcal{X})$. To learn $G$, we propose to train a non-traditional *autoencoder* $\psi_{\theta_G}$ that

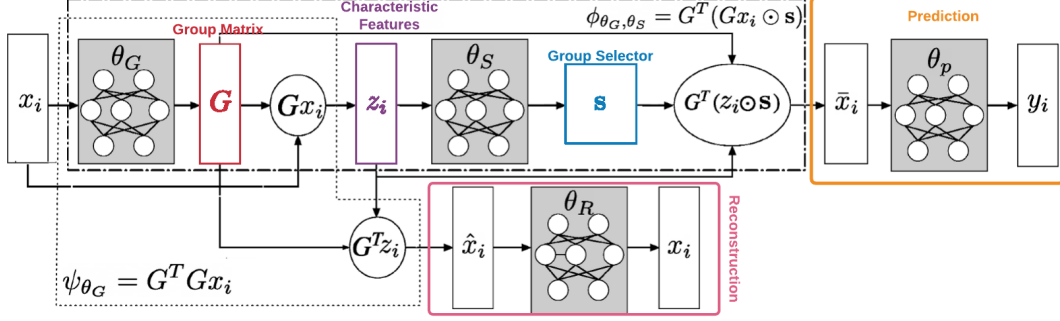

Figure 1: Flowchart of our instance-wise feature grouping framework.

maps the data $\mathbf{X}$ into a low dimensional embedding $\mathbf{Z} \in \mathbb{R}^{n \times k}$ with $\hat{\mathbf{X}} \in \mathbb{R}^{n \times d}$ as its decoded output where $\psi_{\theta_G}(\mathbf{X}) = \hat{\mathbf{X}}$. The *encoder* and *decoder* functions are denoted as $T_G : \mathbb{R}^d \to \mathbb{R}^k$ and $T_G^+ : \mathbb{R}^k \to \mathbb{R}^d$ where for a given sample $i$, $z_i = T_G(x_i) = Gx_i$, $\hat{x}_i = T_G^+(z_i) = G^T z_i$, and $\hat{x}_i = \psi_{\theta_G}(x_i) = T_G^+ \circ T_G(x_i) = G^T G x_i$. Note that each feature of $z_i$ is a summation of only features of the same group, therefore, each feature encapsulates the characteristics of its corresponding group; accordingly, we refer to them as *characteristic features*.

**The Group Selector s.** Each $G$ matrix is coupled with its own $m$-hot vector $\mathbf{s}$ that indicates the $m$ most important groups. By defining the random variable $\mathcal{S} \in \{\mathbf{s} \in \{0,1\}^k || \mathbf{s}| = m\}$, we also learn $\mathbf{s}$ indirectly by learning the distribution $P(\mathcal{S}|\mathbf{Z})$, where $\mathbf{s} = \arg\max_{\mathbf{S}} P(\mathcal{S}|\mathbf{Z})$. This is accomplished given a 2nd autoencoder $\phi_{\theta_S, \theta_G}(x_i) = G^T(Gx_i \odot \mathbf{s})$ which selects $m$ *characteristic features* that corresponds to the $m$ most important groups, where $\odot$ is an element wise product or Hadamard product. Hence, given $\mathbf{X}$ we have $\phi_{\theta_S, \theta_G}(\mathbf{X}) = \bar{\mathbf{X}}$ where the $i_{th}$ row is $\bar{x}_i$.

**Defining Feature Redundancy.** Intuitively, features can be redundant if it is highly dependent on another set of features, we call this *Representation Redundancy*. Simultaneously, features can also be redundant if their inclusion does not improve the data/label dependency, i.e., given the occurrence of a feature, additional features may not provide any extra label-predicting information; we call this *Relevant Redundancy*. Formally, let $\mathcal{X}_j$ be a random variable representing the $j^{\text{th}}$ feature, and let $\mathcal{X} = \{X_1, \ldots, X_d\}$ be a set of all features with the cardinality of $|\mathcal{X}|$. By leveraging mutual information (MI, $I$), we define the two redundancies below.

**Definition 1.** *Feature $X_j$ is Representation Redundant with respect to a set of random variables $\mathbf{Z}$ iff*

$$I(X_j; \mathcal{X}) \neq 0 \quad \text{and} \quad I(X_j; \mathcal{X}|\mathbf{Z}) = 0. \tag{1}$$

**Definition 2.** *Feature $X_j$ is Relevant Redundant with respect to a set of random variables $\mathbf{Z}$ iff*

$$I(X_j; \mathbf{Y}) \neq 0 \quad \text{and} \quad I(X_j; \mathbf{Y}|\mathbf{Z}) = 0. \tag{2}$$

Note that in Def. (1), while condition $I(X_j; \mathcal{X}) \neq 0$ is always true since $X_j \in \mathcal{X}$, it is nevertheless included to preserve the symmetry with Def. (2). Following these definitions, we present our method, the Group Interpreter (gI), which implicitly learns $G$ and $\mathbf{s}$ by maximizing

$$\max_{\theta_G, \theta_S} I(\hat{\mathbf{X}}; \mathbf{X}) + \lambda I(\bar{\mathbf{X}}; \mathbf{Y}), \quad \text{s.t:} \quad \hat{\mathbf{X}} = \psi_{\theta_G}(\mathbf{X}), \quad \bar{\mathbf{X}} = \phi_{\theta_S, \theta_G}(\mathbf{X}). \tag{3}$$

This objective is theoretically motivated by Defs. 1 and 2. Indeed, the $\hat{\mathbf{X}}$ that maximizes $I(\hat{\mathbf{X}}; \mathbf{X})$ captures *Representation Redundancy* while $I(\bar{\mathbf{X}}; \mathbf{Y})$ identifies the optimal $\bar{\mathbf{X}}$ to capture *Relevant Redundancy*. The control parameter $\lambda$ then balances the two criteria. We formally prove these claims in the following two theorems with their proof included in App. C.

**Theorem 1.** *The maximum mutual information $I(\hat{\mathbf{X}}; \mathbf{X})$ is achieved if and only if its characteristic features $\mathbf{Z}$ induced by the model makes $\mathbf{X}$ representative redundant based on Def. (1), i.e.*

$$\max_G I(\hat{\mathbf{X}}; \mathbf{X}) = I(\mathbf{X}; \mathbf{X}) \iff \min_G I(\mathbf{X}; \mathbf{X}|\mathbf{Z}) = 0,$$

$$\text{s.t. } G \in \{0,1\}^{k \times d}, \sum_{i=1}^{k} G_{ij} = 1, \mathbf{Z} = T_G(\mathbf{X}), \hat{\mathbf{X}} = \psi_{\theta_G}(\mathbf{X}). \tag{4}$$

**Theorem 2.** *The maximum mutual information $I(\bar{\mathbf{X}}; \mathbf{X})$ is achieved if and only if its $m$-selected characteristic features $\mathbf{Z} \odot \mathbf{s}$ induced by the model makes $\mathbf{X}$ relevant redundant based on Def. (2), i.e.*

$$\max_{G} I(\bar{\mathbf{X}}; \mathbf{Y}) = I(\mathbf{X}; \mathbf{Y}) \iff \min_{G} I(\mathbf{X}; \mathbf{Y} | \mathbf{Z} \odot \mathbf{s}) = 0,$$

$$\text{s.t. } G \in \{0,1\}^{k \times d}, \sum_{i=1}^{k} G_{ij} = 1, \mathbf{Z} = T_G(\mathbf{X}), \ \bar{\mathbf{X}} = \phi_{\theta_S, \theta_G}(\mathbf{X}), \mathbf{s} \in \{0,1\}^k, |\mathbf{s}| = m. \tag{5}$$

**Approximating Mutual Information.** Since the various distributions required to compute MI are difficult to obtain, we instead maximize MI's variational lower bound [11] as a surrogate. We provide here a summary of the key formulations while leaving the detail derivations to App. G. First, we solve Eq. (3) by first simplifying it into expectations

$$\max_{\theta_G, \theta_S} \quad E_{\mathbf{X}, \hat{\mathbf{X}}}[\log(P(\mathbf{X}|\hat{\mathbf{X}})] + \lambda E_{\mathbf{Y}, \bar{\mathbf{X}}}[\log(P(\mathbf{Y}|\bar{\mathbf{X}})] \quad \text{s.t: } \hat{\mathbf{X}} = \psi_{\theta_G}(\mathbf{X}), \ \bar{\mathbf{X}} = \phi_{\theta_S, \theta_G}(\mathbf{X}) \tag{6}$$

This objective can be approximated by computing its empirical estimate using samples from $P(\mathbf{X}|\hat{\mathbf{X}})$ and $P(\mathbf{Y}|\bar{\mathbf{X}})$. We generate $\hat{\mathbf{X}}, \bar{\mathbf{X}}$ samples via ancestral sampling [26] from

$$P(\hat{\mathbf{X}}|\mathbf{Z}, \mathcal{G}) P(\mathbf{Z}|\mathcal{G}, \mathbf{X}) \ \underline{P(\mathcal{G}|\mathbf{X})} \ P(\mathbf{X}), \tag{7}$$

$$P(\bar{\mathbf{X}}|\mathbf{Z} \odot \mathbf{s}, \mathcal{G}) P(\mathbf{Z} \odot \mathbf{s}|\mathcal{S}, \mathbf{Z}) \underline{P(\mathcal{S}|\mathbf{Z})} P(\mathbf{Z}|\mathcal{G}, \mathbf{X}) \underline{P(\mathcal{G}|\mathbf{X})} P(\mathbf{X}). \tag{8}$$

However, since both $P(\mathbf{X}|\hat{\mathbf{X}})$ and $P(\mathbf{Y}|\bar{\mathbf{X}})$ are unknown, we further use their variational lower bound to approximate their distributions via two additional networks. Specifically, we use $Q_{\theta_R}(\mathbf{X}|\psi_{\theta_G}(\mathbf{X}))$ to approximate $P(\mathbf{X}|\hat{\mathbf{X}})$, and $Q_{\theta_P}(\mathbf{Y}|\phi_{\theta_S, \theta_G}(\mathbf{X}))$ for $P(\mathbf{Y}|\bar{\mathbf{X}})$. This affords us the advantage of combining the four networks ($\psi_{\theta_G}, \phi_{\theta_S, \theta_G}, Q_{\theta_R}$, and $Q_{\theta_P}$) into a large single network and jointly optimize them via Stochastic Gradient Descent (SGD). The resulting formulation becomes

$$\min_{\theta_G, \theta_S, \theta_P, \theta_R} \quad \sum_{i=1}^{n} ||x_i - Q_{\theta_R}(\psi_{\theta_G}(x_i))||^2 - \lambda \sum_{i=1}^{n} p(y_i) \log(Q_{\theta_P}(y_i|(\phi_{\theta_S, \theta_G}(x_i))). \tag{9}$$

Solving Eq. (9) relies on drawing samples from $P(\mathcal{G}|\mathbf{X})$ and $P(\mathcal{S}|\mathbf{Z})$. However, since $G$ and $\mathbf{s}$ are constrained to be indicators, how do we enforce the categorical constraint on the output of $\psi_{\theta_G}$ and $\phi_{\theta_S, \theta_G}$? We clarify how adding a Gumbel-softmax layer [27] achieves this in the next section.

**Gumbel-Softmax.** Standard networks cannot perform backpropagation through samples. Gumbel-softmax overcome this obstacle by generating differentiable samples from a categorical distribution. Leveraging this technique, we sample a $k$-dimensional vector $\epsilon$ from a Gumbel distribution where its $i$th element is sampled via $\epsilon_i = -\log(-\log u_i), u_i \sim \text{Uniform}(0,1)$. This enables us to apply the reparameterization trick [28], which consequently samples from a concrete distribution, $C \sim \text{Concrete}(\log p_1, ..., \log p_k)$, where the $i$th element is computed with

$$C_i = \frac{\exp(\log p_i + \epsilon_i)/\tau}{\sum_{j=1}^{k} \exp(\log p_j + \epsilon_j)/\tau} \quad \text{s.t.} \lim_{\tau \to 0} P(C_i = 1) = \frac{p_i}{\sum_{j=1}^{k} p_j}. \tag{10}$$

The sharpness of the concrete distribution is controlled by $\tau$; where as $\tau \to 0$, the concrete random variable approaches to the categorical distribution as defined in Eq. (10). Therefore, $\theta_G$ in Fig. 1 represents the combination of a network $Q_{\theta_G} : \mathbb{R}^d \mapsto \mathbb{R}^{k \times d}$ with a Gumbel-softmax layer. Since $Q_{\theta_G}$ outputs a dimension of $k \times d$, the output can be reorganized into $d$ columns of size $k$ vectors, where the $i$th column represents the group membership probability $[p_1, ..., p_k]^T$ for the $i$th feature. By passing each column into the Gumbel-softmax layer, it consequently generates a one-hot vector for each column of the $G$ matrix, representing samples from $P(\mathcal{G}|\mathbf{X})$. Similarly, $\theta_S$ consists of $Q_{\theta_S} : \mathbb{R}^k \mapsto \mathbb{R}^k$ with a Gumbel-softmax layer. However, an $m$-hot vector is generated by repeating Gumbel-softmax $m$ times. Specifically, let each trial be $C^t$, then $\mathbf{s}$ is generated by

$$C^t \sim \text{Concrete}(Q_{\theta_S}), \text{ for } t = 1, \ldots m, \quad \mathbf{s} = [\mathbf{s}_1, \ldots, \mathbf{s}_k]^T, \quad \mathbf{s}_j = \max_t C_j^t. \tag{11}$$

**Discovering the Number of Groups.** Instead of randomly guessing the number of groups, $k$, is there a theoretical guideline? We tackle this question from an information-theoretic perspective, by asking if there exists a minimum $k$ such that all relevant information is preserved. To state the question precisely, what is the minimum $m$ and $k$ such that $\mathrm{I}(\mathbf{X}; \mathbf{Y}) = \mathrm{I}(\phi_{\theta_S, \theta_G}(\mathbf{X}); \mathbf{Y})$?

Since we compress the original features into characteristics features and then remove the least important groups, how can information retention be possible? By studying the simpler case where all groups are kept, we identified a set of conditions which this becomes possible and discovered a lower bound for $k$. Specifically, we simplify the problem by letting $m = k$ such that $\psi_{\theta_G} = \phi_{\theta_S, \theta_G}$, then we study if $T_G^+$ and $T_G$ individually preserves information. Conceptually, since $\psi_{\theta_G} = T_G^+ \circ T_G$, information is preserved if $T_G^+$ and $T_G$ both preserve information. This intuition is supported by Kraskov et al. [29]: they show that MI is invariant under diffeomorphism mappings. Therefore, we investigate if $T_G$ and $T_G^+$ are diffeomorphisms and formalize these findings in the following two lemmas with their proof in App. B.

**Lemma 1.** *The decoder $T_G^+ : \mathbf{Z} \to \mathrm{Im}(T_G^+)$ is a Diffeomorphism map.*

**Lemma 2.** *If $k < d$ then the mapping $T_G : R^d \to R^k$ is not injective, thus **not** a diffeomorphism.*

Since $k$ is always less than $d$ when features are grouped together, our analysis proves that $\psi_{\theta_G}$ cannot be a diffeomorphism; a disappointing result. Yet, we note that while having diffeomorphism guarantees information preservation, nothing is stated about non-diffeomorphism mappings. Indeed, by digging deeper, we found that information preservation is still possible under certain non-diffeomorphism conditions. Specifically, we prove that relevant information can still be preserved if $k$ is sufficiently large, i.e., larger than the number as defined by Eq. (63). In fact, in these cases, we proved the existence of a matrix $G$ such that $I(\hat{\mathbf{X}}; \mathbf{Y}) = I(\mathbf{X}; \mathbf{Y})$. We formally state this finding in Theorem 3; the proof can be found in App. D.

**Theorem 3.** *Let $\mathcal{X} = \{X_1, \ldots, X_d\}$ be a random variable that consists of all features , let relevant features $\mathcal{U}$ be*

$$\mathcal{U} = \{X_j \mid I(X_j; \mathbf{Y}) \neq 0 \vee \exists \mathcal{A} \subseteq \mathcal{X}\ I(X_j; \mathbf{Y} | \mathcal{A}) \neq 0\}, \tag{12}$$

*and let irrelevant features be $\mathcal{U}^c$, then, $\exists G \in \mathcal{G}$ such that $I(T_G(\mathbf{X}); \mathbf{Y}) = I(\mathbf{X}; \mathbf{Y})$ if*

$$k \geq |\mathcal{U}| + \mathbb{1}(|\mathcal{U}| \neq d) \tag{13}$$

*where $\mathbb{1}(|\mathcal{U}| \neq d)$ is an indicator function equal to one when $|\mathcal{U}| \neq d$ and zero when $|\mathcal{U}| = d$.*

While Theorem 3 provides a theoretical bound for $k$, in practice, the computation of $\mathcal{U}$ assumes prior access to complex posterior distributions. Since this assumption is rarely true in practice, we provide an alternative bound that only requires the correlation coefficient between the features and the labels. We formally state this theorem and its corollaries below with their proofs in App. E.

**Theorem 4.** *Given $\rho$ as the correlation measure and $\mathcal{C} = \{X_j | \rho(X_j; \mathbf{Y}) \neq 0 \vee \exists \mathcal{A}\ \rho(X_j; \mathbf{Y} | \mathcal{A}) \neq 0\}$ then $|\mathcal{C}| \leq |\mathcal{U}|$.*

**Corollary 4.1.** *Theorems 3 and 4 yields a lower bound for $k$ where $|\mathcal{C}| + \mathbb{1}(|\mathcal{C}| \neq d) \leq k$.*

**Corollary 4.2.** *For Gaussian distributions the inequality turns into equality where $|\mathcal{C}| = |\mathcal{U}|$.*

By leveraging Corollary 4.1, a more tractable set $\mathcal{C}$ can be obtained in place of $\mathcal{U}$ to bound $k$.

**Computational and Memory Complexities.** Since our algorithm can be solved via SGD, gI has efficient memory and computational complexities of $O(kd^2)$ and $O(nkd^2)$ respectively. For a detailed derivation of these complexities, refer to App. H.

**Feature Selection vs Explaining Black-Box Models.** Due to the common confusion between feature selection, and black-box explanatory models (BEM), we emphasize that our focus is feature selection. Our method, gI, learns a classifier $Q_{\theta_p}(y|\phi_{\theta_S, \theta_G}(x))$ via feature grouping that approximates the true underlying posterior $P(\mathbf{Y}|\mathbf{X})$. Note that one can easily extend gI to explain black-box models by changing $P(\mathbf{Y}|\mathbf{X})$ to a complex black-box learned classifier $P_M(\mathbf{Y}|\mathbf{X})$ (e.g., neural networks [30], random forest [31]) similar to Chen et al. [11]; where, $Q_{\theta_p}(y|\phi_{\theta_S, \theta_G}(x))$ now approximates $P_M(\mathbf{Y}|\mathbf{X})$ by learning from training data with $\mathbf{Y}$ generated from the output of $P_M(\mathbf{Y}|\mathbf{X})$ for each $x_i$. Although gI can be easily extended to BEM, we leave this extension for future research.

## 3 Experiments

**Datasets.**  We validate the theoretical claims with nine synthetic datasets constructed from a combination of three *Representation* $(D_1, D_2, D_3)$ and three *Relevance* $(R_1, R_2, R_3)$ redundancy patterns as shown in Table 1. Recall that $X_j$ indicates the $j$th feature. For *Representation Redundancy* ($D$ patterns), the features within the same parentheses are correlated with each other. For *Relevance Redundancy* ($R$ patterns) the $P(\mathbf{Y} = 1|\mathbf{X})$ is directly proportional to a function of the features indicated. We generate 100000 training, 1000 validation, and 1000 test samples for each combination. For each combination, we evaluate gI's ability to correctly identify the number of groups ($k$), the redundancy patterns, and classification results.

We also evaluate our method on a real-world gene expression data as quantified by RNA sequencing from the COPDGene Study, an observational study to identify genomic markers associated with chronic obstructive pulmonary disease (COPD) [32]. The dataset is divided into a training and test set of 1500 and 407 patients along with the expression of 439 most relevant genes based on Gene Ontology categories [33]. We additionally test on benchmark image datasets from MNIST, and Fashion MNIST (F-MNIST) [34, 35] to evaluate our method's ability to generate visual results.

| | |
|---|---|
| $D_1$ | $(X_1, X_2), (X_3, X_4)$ |
| $D_2$ | $(X_1, X_3), (X_2, X_4)$ |
| $D_3$ | $(X_1, X_3, X_4), (X_2)$ |
| $R_1$ | $P(Y = 1|X) \propto e^{X_1 * X_3}$ |
| $R_2$ | $P(Y = 1|X) \propto e^{\sum_{i=1}^{4} X_i^2 - 4}$ |
| $R_3$ | $P(Y = 1|X) \propto e^{-\sin(2X_1) + 2|X_2| + X_3 + \exp(-X_4 - 2.4)}$ |

Table 1: Synthetic data generation patterns

| Model | MNIST-2 | MNIST-10 | F-MNIST |
|---|---|---|---|
| gI | $96.7 \pm 0.2$ | $\mathbf{91.6 \pm 0.85}$ | $94.6 \pm 0.6$ |
| L2X | $97.1 \pm 0.5$ | $80.5 \pm 2.5$ | $96.0 \pm 0.6$ |
| shap | $\mathbf{99.24 \pm 0.46}$ | $90.8 \pm 1.9$ | $94.45 \pm 2.62$ |
| INV | $91.23 \pm 3.48$ | $77.94 \pm 2.35$ | $89.63 \pm 3.45$ |
| Lasso | $96.01 \pm 0.2$ | $86.03 \pm 0.02$ | $\mathbf{96.7 \pm 0.0}$ |
| Group Cluster | $94.36 \pm 0.05$ | $85.0 \pm 0.09$ | $92.04 \pm 0.06$ |
| OSCAR | $95.56 \pm 0.31$ | $90.94 \pm 0.27$ | $95.0 \pm 0.3$ |
| OWL | $95.8 \pm 0.25$ | $90.92 \pm 0.31$ | $94.90 \pm 0.16$ |
| LPA | $95.63 \pm 0.56$ | $87.72 \pm 1.23$ | $94.83 \pm 0.97$ |

Table 2: gI $m = 1$, $k = 2$, image Classification accuracy comparison.

**Experimental Settings.**  All experimental accuracies are reported via the mean and standard deviation of 10 runs. The experiments are implemented with Python, Numpy, Sklearn, and TensorFlow [36, 37, 38, 39] on a single NVIDIA GTX 1060Ti GPU. We use a neural network of width 100 and depth 2 to generate the probability inputs for the Gumbel-Softmax to obtain $G$ and $S$; the Gumbel temperature was set to 0.1. ReLU was used as the activation function with softmax at the final layer for prediction. Adam optimizer with a learning rate of 0.001 and hyperparameters $\beta_1 = 0.9, \beta_2 = 0.999$ was used without further tuning. All datasets are centered to 0 and normalized to have a standard deviation of 1. For all data, we used two fully connected layers of width 32 and 16. All $\lambda$s are identified by maximizing the objective given a validation set.

**Competing Methods.**  We compare gI against nine related feature selection and explainable methods. For all methods, we learn from samples of the true underlying posterior $P(\mathbf{Y}|\mathbf{X})$ (i.e., ground-truth training data) to fairly compare them.

- **Global feature selection**: **Lasso** (Least Absolute Shrinkage and Selection Operator) [23] is a regression method that utilizes $l_1$ regularization to induce sparsity and effectively perform feature selection. **GLasso** (sparse Group Lasso) [40, 41] is a Lasso version that assumes a feature grouping structure, enforces $l_1$ sparsity and performs group selection with an $l_{1,2}$ regularizer.
- **Deep instance-wise feature selection**: **SHAP** (SHapley Additive exPlanations) [12] provides a unified framework for explaining models by identifying a class of additive feature importance measures for prediction. SHAP learns feature importance (Shapley values) based on a game theoretic approach. **L2X** [11] performs instance-wise feature selection for explaining black-box models by maximizing the mutual information between the selected features and the response variable. In addition, L2X uses Gumbel softmax to learn a continuous relaxation of the feature selector. **INV** (INVASE) [14] is an extension over L2X without the need to specify the number of selected features in advance and is capable of discovering subsets of features with a different size per instance. **LPA** (Learn to Pay Attention) [15] is A visual-attention based deep learning model for learning saliency maps from the original input images. We adapted the original model to a fully-connected version based on the architecture used by all methods in this paper for fair comparison. Note that these models cannot learn and do not use the feature grouping structure.

- **CAE** [42]: An end-to-end unsupervised global feature selection to reconstruct the input data, with a Gumbel softmax layer as the encoder and a standard neural network as the decoder. As an unsupervised method, we only apply CAE to the visual MNIST and F-MNIST experiments.
- **Global feature selection with group learning: OSCAR** (octagonal shrinkage and clustering algorithm for regression) [18] learns feature groups in regression by regularizing the weights with $l_1$ and pairwise $l_\infty$ norm to encourage correlated predictors that have a similar effect on the response to form clusters represented by the same coefficient. **OWL-Lasso** [43] performs linear regression and group feature selection by utilizing a weighted $l_1$ regularization. **Group Cluster** groups the features based on hierarchical correlation clustering [44] followed by GLasso.

**Results on Synthetic Data.** We use synthetic datasets to answer the following questions:
- Can our model correctly identify the features that are highly dependent on each other?
- Can our model correctly identify the most relevant features in predicting **Y**?
- Is $k$ based on Theorems 3 and 4 a tight lower bound?
- How does the accuracy of our method compare to existing interpretable methods?

Given all 9 redundancy combinations of $(D_i, R_j)$ plus six additional Gaussian noise features, Table 3 indicates that both gI$(m = k)$ and gI$(m < k)$ are capable of achieving high class accuracy while learning the latent group structure (high representation (rep) and relevant (rel) accuracies), thereby confirming Thms. 1 and 2. Moreover, since the recommended $k$ value by Thm. 4 is a lower bound, we investigated the bound by plotting the classification accuracy at each increment of $k$ in Fig. 2 and circle the lower bound predicted by Thm. 4. As predicted by our theorem, after the number of groups passes the lower bound calculated by Thm. 4 the preservation of the mutual information between **X** and **Y** is possible and indeed after the number of groups passes the lower bound there is no decline in the classification accuracy.

| | Class Acc (**gI**($m = k$)) | | | $k$ (**gI**($m=k$)) | | | Group Rep Acc (**gI**($m = k$)) | | | Class Acc (**gI**) | | | $m$ (**gI**) | | | Group Rel Acc (**gI**) | | |
|---|---|---|---|---|---|---|---|---|---|---|---|---|---|---|---|---|---|---|
| | $D_1$ | $D_2$ | $D_3$ | $D_1$ | $D_2$ | $D_3$ | $D_1$ | $D_2$ | $D_3$ | $D_1$ | $D_2$ | $D_3$ | $D_1$ | $D_2$ | $D_3$ | $D_1$ | $D_2$ | $D_3$ |
| $R_1$ | $96.9 \pm 1.5$ | $100 \pm 0$ | $99.5 \pm 1.5$ | 3 | 2 | 2 | $99.8 \pm 0.3$ | $100 \pm 0$ | $100 \pm 0$ | $91.0 \pm 3$ | $99.6 \pm 1$ | $98.5 \pm 2$ | 1 | 1 | 1 | $100 \pm 0$ | $100 \pm 0$ | $100 \pm 0$ |
| $R_2$ | $100 \pm 0$ | $100 \pm 0$ | $99.6 \pm 0.4$ | 3 | 3 | 3 | $100 \pm 0$ | $100 \pm 0$ | $99 \pm 1.8$ | $92 \pm 3$ | $95.4 \pm 2$ | $94 \pm 1$ | 2 | 2 | 2 | $100 \pm 0$ | $100 \pm 0$ | $100 \pm 0$ |
| $R_3$ | $98.9 \pm 0.8$ | $97.6 \pm 0.6$ | $99.2 \pm 0.8$ | 3 | 3 | 3 | $100 \pm 0$ | $100 \pm 0$ | $99.5 \pm 0.9$ | $94.4 \pm 3$ | $94.7 \pm 2$ | $90.9 \pm 5$ | 2 | 2 | 2 | $100 \pm 0$ | $100 \pm 0$ | $100 \pm 0$ |

Table 3: Measuring gI's ability to identify the most relevant groups using nine redundancy patterns. Note that gI is capable of identifying the relevant groups while achieving a high classification accuracy.

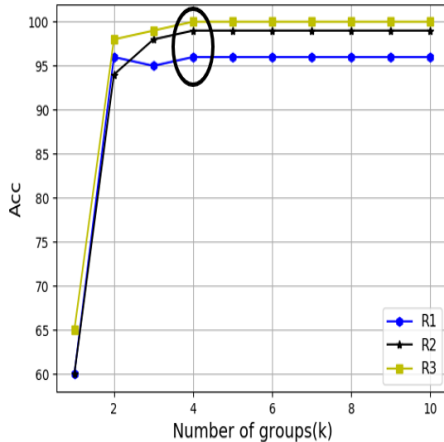

Figure 2: Accuracy versus number of groups used: We circle the number of groups predicted by Thm. 4.

| | Data | $D_1$ | $D_2$ | $D_3$ | $D_1 + D_2$ |
|---|---|---|---|---|---|
| **gI** | $R_1$ | $\mathbf{98.4 \pm 1}$ | $99.7 \pm 0.46$ | $95 \pm 3.8$ | $\mathbf{98.7 \pm 1.3}$ |
| | $R_2$ | $100 \pm 0$ | $\mathbf{99.5 \pm 0.5}$ | $100 \pm 0$ | $100 \pm 0$ |
| | $R_3$ | $\mathbf{98.8 \pm 0.9}$ | $\mathbf{99.4 \pm 0.6}$ | $\mathbf{99.4 \pm 0.6}$ | $\mathbf{99.2 \pm 0.4}$ |
| **L2X** | $R_1$ | $85 \pm 3.5$ | $\mathbf{100 \pm 0.0}$ | $85.7 \pm 6$ | $88 \pm 4$ |
| | $R_2$ | $95 \pm 2$ | $95 \pm 1.4$ | $95 \pm 2.1$ | $99.7 \pm 0.5$ |
| | $R_3$ | $94 \pm 2.2$ | $95 \pm 1.1$ | $87.7 \pm 1$ | $93 \pm 1.3$ |
| **Shap** | $R_1$ | $70.25, \pm 0.83$ | $\mathbf{100 \pm 0.0}$ | $\mathbf{100 \pm 0.0}$ | $89.2 \pm 0.97$ |
| | $R_2$ | $88.0 \pm 0.0$ | $94 \pm 0.0$ | $82 \pm 0.0$ | $94.4 \pm 0.48$ |
| | $R_3$ | $94.6 \pm 0.48$ | $95 \pm 0.0$ | $95 \pm 0.0$ | $95.4 \pm 0.48$ |
| **INV** | $R_1$ | $87.2 \pm 3$ | $88.5 \pm 3$ | $87.2 \pm 3$ | $86 \pm 2$ |
| | $R_2$ | $73 \pm 3$ | $80.9 \pm 4$ | $68 \pm 3.5$ | $75 \pm 4$ |
| | $R_3$ | $74 \pm 4$ | $79 \pm 2$ | $73 \pm 4$ | $74 \pm 4$ |
| **Lasso** | $R_1$ | $49 \pm 3$ | $\mathbf{100 \pm 0.0}$ | $\mathbf{100 \pm 0.0}$ | $74 \pm 1$ |
| | $R_2$ | $66 \pm 1$ | $61 \pm 1$ | $67 \pm 2$ | $58 \pm 2$ |
| | $R_3$ | $75 \pm 2$ | $84 \pm 2$ | $59 \pm 3$ | $81 \pm 8$ |
| **GLasso** | $R_1$ | $49 \pm 1$ | $\mathbf{100 \pm 0.0}$ | $\mathbf{100 \pm 0.0}$ | $76.0 \pm 0.4$ |
| | $R_2$ | $64 \pm 0.08$ | $62 \pm 0.4$ | $49 \pm 0.4$ | $56 \pm 0.4$ |
| | $R_3$ | $74 \pm 1$ | $83 \pm 0.5$ | $68 \pm 1.3$ | $79 \pm 2$ |
| **OSCAR** | $R_1$ | $49.0 \pm 0.31$ | $\mathbf{100 \pm 0.0}$ | $\mathbf{100 \pm 0.0}$ | $50.0 \pm 0.0$ |
| | $R_2$ | $50.0 \pm 0.3$ | $50.3 \pm 0.11$ | $50.0 \pm 0.3$ | $50.1 \pm 0.05$ |
| | $R_3$ | $74.7 \pm 0.2$ | $84.03 \pm .15$ | $66 \pm 0.14$ | $79.2 \pm 0.13$ |
| **OWL** | $R_1$ | $49.0 \pm 0.31$ | $\mathbf{100 \pm 0.0}$ | $\mathbf{100 \pm 0.0}$ | $50.0 \pm 0.0$ |
| | $R_2$ | $50.0 \pm 0.3$ | $50.3 \pm 0.11$ | $50.0 \pm 0.3$ | $50.1 \pm 0.05$ |
| | $R_3$ | $74.7 \pm 0.2$ | $84.03 \pm .15$ | $66 \pm 0.14$ | $79.2 \pm 0.13$ |

Table 4: The classification prediction accuracy on synthetic datasets.

In Table 4, we compare gI against competing methods. In addition to mixing $D$ and $R$ redundancies together, we increase the data complexity by combining $D_1$ relationships with $D_2$ as $D_1 + D_2$, where half of the samples generated are randomly chosen to have $D_1$ redundancies while the other half is set to have $D_2$ redundancies. Since only our model performs classification based on instance-wise

grouping of features, the $D_1 + D_2$ pattern is of particular interest to validate our advantage, i.e., given its correct assumption of the data, our model is expected to outperform all alternative methods. By marking the best results as bold in Table 4, we can see that gI is almost always the best performing classifier. As expected, the accuracy difference is particularly prominent with $D_1 + D_2$.

**COPDGene Dataset.** Given the effect of smoking on health, there is significant interest in its impact on gene expression. Specifically, how does exposure alter gene expression, and how do groups of genes exhibit *coordinated* changes given exposure? This data highlights the insufficiency of learning a single *global* group structure because smokers and non-smokers may be characterized by completely different gene groups. We emphasize the importance of identifying this variability by applying gI to the COPDGene dataset to learn the most predictive group of genes on smoking status. Instead of trying to pinpoint a *single group* of the most important genes, gI's instance-wise capability is designed to automatically identify *multiple groups*.

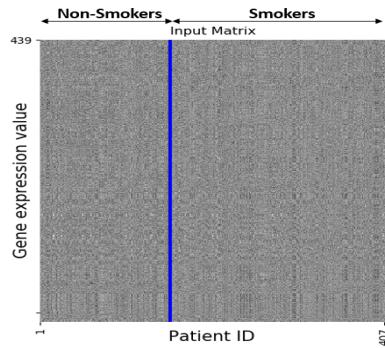

Figure 3: Gene expression (input features) of patients $X^T$. No pattern is visually noticeable.

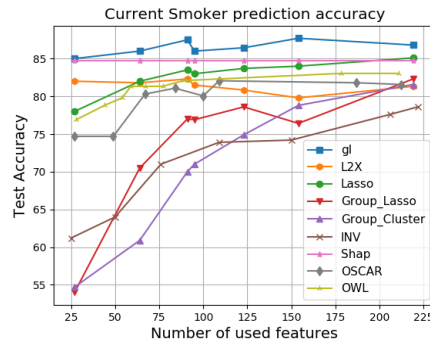

Figure 4: Prediction accuracy vs. number of features selected. gI consistently outperforms other methods.

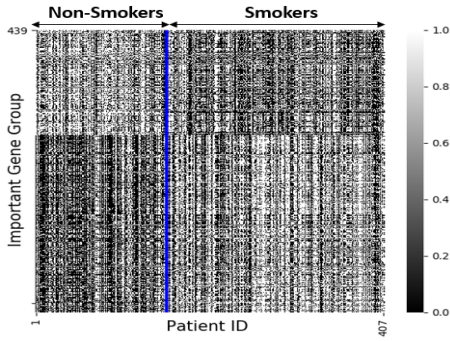

Figure 5: The important genes selected by $G$ and $\mathbf{s}$. The selected genes (rows) are indicated by *white* pixels for each patient (column) and *black* when not selected.

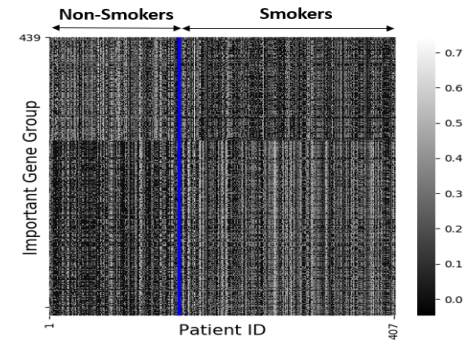

Figure 6: Each column represents the characteristic features of each patient, i.e., $\mathbf{Zs}$. Note that visually, smokers and non-smokers are clustered appropriately.

Fig. 4 compares the test accuracy between several competing methods given increasing number of features; *gI consistently achieves the highest accuracy*. Moreover, note that *Group Lasso* represents the traditional method of applying biologically predefined groups. Yet, even when domain knowledge is incorporated within *Group Lasso*, the instance-wise capability of gI identifies the gene groups that achieves much higher predictive accuracy.

We next studied the group structure produced by gI. First, notice that the original gene expression matrix in Fig. 3 lacked any visually noticeable patterns. We then plot the most relevant group of genes selected by $G$ and $s$ in Fig. 5, where the selected genes (rows) are indicated by the *white* pixels for each patient (column). Even with the high variance between patients, a pattern emerges; gI has identified the group of genes that are common across smokers and non-smokers respectively. As suggested by our results, there exists a visual difference in gene expression between the two groups and gI has identified the specific genes for each group. We next plot out the *characteristic features*

formed by each $G$ matrix. As predicted by Thm. 4, this compressed representation of the original input features retained the most relevant information despite the compression.

Since different genes tended to be selected in smokers compared to nonsmokers, we performed Gene Set Enrichment Analysis (GSEA) as implemented in the GenePattern Cloud instance (https://cloud.genepattern.org/) using a set of curated immunologic gene signatures (the C7 set) from the Molecular Signatures Database. Immunologic signatures is well suited for analysis of blood expression data since the majority of cells present in blood are immune cells. The analysis determines whether predefined gene sets are enriched in the extremes of the ranked list of genes, where ranking is based on each gene's likelihood of being selected among each of the two cohorts. In this analysis, using a 10 percent false discovery rate, 20 significantly enriched immunologic gene sets were identified among the most frequently selected genes for smokers, whereas no similar enrichment of immunologic signatures was observed among the genes selected the most among nonsmokers.

**Competing Method Performance on Image Datasets.** Table 2 reports the classification accuracy for all methods on MNIST-2, MNIST-10 (all 10 digits) and F-MNIST. Notice that while L2X and Shap performed slightly better on the simpler F-MNIST and MNIST-2 (3 vs. 8) datasets, gI performed better on the more complex MNIST-10 (all 10 digits) dataset.

In Fig. 7, we compare the visual patterns generated by gI against several best performing deep models. For each image, each method identifies and displays the most informative pixel group in *white*; the top row is the original image while the results of each method are displayed below. While L2X, LPA, and Shap are all instance-wise and can achieve high predictive accuracy, it is not clear visually from their white pixels in Fig. 7 why these pixels are important. CAE outputs a discernible shape of 8, however, its features are global, resulting in the same pixel choice across all samples. In contrast, gI discovered the pixels that are equally important, resulting in a visually compelling segmentation in the shape of the classification object. Our result suggests that capturing and identifying redundancies within the data produces visually interpretable explanations, highlighting the importance of combining group structure with instance-wise flexibility. While other methods struggle to identify the different digits and clothing, gI handled the complexity independent of the number of classes.

An even more challenging task is to also capture the style variation within the same class. We highlight this ability with 10 digits of diverse shapes in Fig. 7 under INSTANCE-WISE MNIST STYLE; a larger collection showcasing a variety of style variation results can be found in App. F. For these results, notice how the explanatory pixels follow closely to the style of the original image.

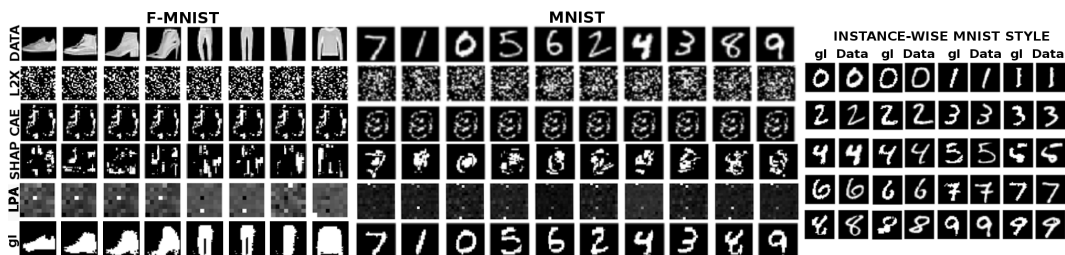

Figure 7: Comparing the most important pixels as identified by each competing algorithm.

## 4 Conclusion

Our theoretical contribution formally defines the concept of redundancy between features based on MI. This clarifies how features can be grouped together, and how many groups should exist while retaining the most relevant information. It further enables us to formulate an objective (gI) that captures these redundancies on an instance-wise basis. Our theories are corroborated by both synthetic and real experimental results. We have applied our instance-wise feature group discovery and selection method to lung disease gene expression data; of which we discovered gene expression patterns common to smokers and non-smokers respectively.

# 5 Broader Impacts

In this paper, we introduce a novel algorithm for instance-wise feature group discovery and selection. The algorithm learns mapping functions that identify the appropriate group membership of each feature along with each group's importance as an instance-wise label predictor. Namely, we have focused our paper on feature selection to model the features important for capturing the information in the underlying true posterior $P(\mathbf{Y}|\mathbf{X})$. While we have focused on feature selection, there are also other strategies to define and approach interpretability [45].

Instead of estimating the posterior distribution $P(\mathbf{Y}|\mathbf{X})$, one can apply our method to capture the information for trained black-box models $P_M(\mathbf{Y}|\mathbf{X})$, e.g., deep neural networks and random forests. Consequently, the algorithm can be used to perform instance-wise group feature selection on the black-box model, learning the features which a given black-box model perceives as important. In this approach to explainability, our method has the potential impact on making black-box models explainable in terms of knowing how the features were used during prediction. This gives rise to future research directions that can help data scientists check for bias, fairness, vulnerabilities of the models they use [46, 47].

Although this paper focuses on the machine learning aspect of our discovery, our work is also relevant from its consequential findings on the lung disease dataset. The feature selection results on the lung disease data allow us to discover the genes that interact together for predicting smoking and non-smoking. This can potentially impact our understanding of lung disease, in particular by identifying cooperative relationship between genes that can delineate important aspects of their biological functions. However, to make an impact to medical research would require further and careful investigation to confirm the findings with appropriate medical collaborators. As a warning to our ML and data analyst colleagues, we encourage applying ML to applications that is beneficial to society, such as health. But, to do so properly, one needs to work closely with domain expert collaborators to make nontrivial contributions to their fields of research.

Beyond applications to lung disease, learning important features for prediction and the features that interact together is important in genetic understanding of other diseases [48, 49]. In general, feature selection has been impactful in a variety of domains beyond medicine – for example, climate [50], law[51]. Given the potential impact it can have, including on the most pressing diseases of today, we seek to widely disseminate this research and make our source code publicly available at `https://github.com/ariahimself/Instance-wise-Feature-Grouping`.

Lastly, while our method is useful in identifying feature groups that interact together for prediction. We caution that this does not imply causation, and poses a potential misuse of our technique. Additionally, since our model is learned from a training set, its conclusions are limited by the quality and characteristics of what it was trained on. Therefore, inherent biases that pre-existed in the data will lead to biased feature groups and conclusions. As with any supervised machine learning algorithms, our method can be applied to a variety of applications (e.g., health, climate, image analysis) with potential impact to multiple sectors of society. Our intent is to build such models for societal good and we encourage others to as well.

## Acknowledgements

The work described was supported in part by Award Numbers U01 HL089897, U01 HL089856, R01 HL124233, and R01 HL147326 from the National Heart, Lung, and Blood Institute, the FDA Center for Tobacco Products (CTP), and NSF IIS 1546428. The authors would also like to thank Stratis Ioannidis, Amirreza Farnoosh,Yale Chang, Davin Hill, and Zulqarnain Khan for helping to review the paper and provide fruitful discussions. Their insightful comments in conjunction with the comments of the anonymous reviewers have helped to improve this paper tremendously.

## Footnotes

*Signifies equal contribution.

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
