[Supplementary Material · appendix.pdf]

# A Mutual information

In this section, we provide a short tutorial on Mutual Information (MI, $I$) and some of its known properties we use within our proofs. Given two random variables $X$ and $Y$, MI measures the distributional distance between $P(X, Y)$ and $P(X)P(Y)$ via the Kullback-Leibler divergence. The equation for $I(X, Y)$ is written as

$$I(X; Y) = D_{\text{KL}}(P_{(X,Y)} \| P_X \otimes P_Y) \tag{14}$$

where Kullback-Leibler divergence of two distributon $P, Q$ is defined as

$$D_{\text{KL}}(P \| Q) = \sum_{x \in \mathcal{X}} P(x) \log \left( \frac{P(x)}{Q(x)} \right). \tag{15}$$

Here, $\mathcal{X}$ is the defined domain of the random variable $X$.

**Property 1**: MI is non negative, or $I(X; Y) \geq 0$.

**Property 2**: MI is symmetric, or $I(X; Y) = I(Y; X)$.

**Property 3**: MI with joint distributions has the following chain rule:

$$I(Y; X_1, \ldots, X_n) = I(Y; X_1) + I(Y; X_2 | X_1) + \cdots + I(Y; X_n | X_1, \ldots, X_{n-1}). \tag{16}$$

**Property 4**: MI of $X$ and $Y$ can be increase or decrease via the conditioning of a third variable $Z$, i.e., it is possible that $I(X; Y | Z) \geq I(X; Y)$ or $I(X; Y | Z) \leq I(X; Y)$ when $Z$ is introduced. As a corollary to this property, it is therefore possible that $I(X; Y) = 0 \nRightarrow I(X; Y | Z) = 0$.

**Property 5**: MI between variables is conserved through diffeomorphism. We formally state this property as Lemma 3 below.

> **Lemma 3.** *Mutual information is invariant under reparametrization of the marginal variables if $X' = F(X)$ and $Y' = G(Y)$ are diffeomorphism, then:*
> $$I(X; Y) = I(F(X); G(Y))$$

**Property 6**: MI cannot be increased via any deterministic transformation of its variables. We formally state this property as Lemma 4 below.

> **Lemma 4.** *According to inequality of data processing for any deterministic transformation $f$ we have:*
> $$I(f(X); Y) \leq I(X; Y).$$

Therefore, as a key corollary used in our proof, we note that
**Corollary 4.3.** *Using the encoder $T_G$ as $f$ will not enhance the mutual information. meaning:*
$$I(T_G(X); Y) \leq I(X; Y).$$

## A.1 A note to mention:

We defined $\mathcal{X} = \{X_1, \ldots, X_d\}$, and $\mathcal{A} \subseteq \mathcal{X}$, where $X_j$ is a random variable representing feature $j$ and $\mathcal{A}$ is a random variable that has a subset of features. But we can also define these notations as follows: $\mathcal{X} = \{(\lambda^1, \ldots, \lambda^d) | \forall j \lambda^j = X_j \lor \lambda^j = 0\}$, and $\mathcal{A} \in \mathcal{X}$. Note that $|\mathcal{X}| = 2^d$, because $\lambda^j$ has two choices, to either reflect the random variable $X_j$ or be 0. Furthermore, the every element in $\mathcal{X}$ has can be seen as a projection of the element that all $\lambda$ are non zero or $(x_1, \ldots, x_d)$. Using the data processing Lemma 4. Hence we can say $I(\mathcal{X}; \mathbf{Y}) = I(X; \mathbf{Y})$.

# B Proof for Lemmas 1 and 2

**Lemma 1.** *The decoder $T_G^+ : \mathbf{Z} \to \text{Im}(T_G^+)$ is a Diffeomorphism map.*

*Proof.* Diffeomorphism map is a map that is smooth and bijective.

**Smoothness**: The decoder $T_G^+$ in a linear map, hence smooth. We need to prove that $T_G^+$ is bijective.

**Injectivity**: First we prove it is injective, and therefore

$$\forall z_i, z_j \in \mathbb{R}^k, \; T_G^+(z_i) = T_G^+(z_j) \rightarrow z_i = z_j, \tag{17}$$

The Group matrix $G \in \mathbb{R}^{k \times d}$ can be represented as its rows which we represent as $\hat{g}_i \in \mathbb{R}^d$, $\forall i = 1, \ldots, k$. because the group are not overlapping, the rows of $G$ are always orthogonal with each other which means:

$$\forall i \neq j, \; 1 \leq i < j \leq k \quad \langle \hat{g}_i, \hat{g}_j \rangle = 0. \tag{18}$$

Let the characteristic features of the $i^{\text{th}}$ sample be represented as $z_i \in \mathbb{R}^k = [z_{i,1}, \ldots, z_{i,k}]^T$, the decoder is defined as $T_G^+(z_i) = G^T \cdot z_i$ which is equal to

$$G^T \cdot z_i = z_{i,1}\hat{g}_1 + \cdots + z_{i,k}\hat{g}_k. \tag{19}$$

Assume $T_G^+(z_i) = T_G^+(z_j)$, we will prove that $z_i = z_j$. First, because $T_G^+$ is linear, we can combine $z_i, z_j$, hence $T_G^+(z_i) = T_G^+(z_j)$ lead to $T_G^+(z_i - z_j) = 0$, using Eq. 19 we obtain

$$T_G^+(z_i - z_j) = (z_{i,1} - z_{j,1})\hat{g}_1 + \cdots + (z_{i,k} - z_{j,k})\hat{g}_k = 0. \tag{20}$$

Note that $(z_{i,\eta} - z_{j,\eta})$ is a scalar value for all $\eta$, therefore, we can multiply $\hat{g}_\eta^T$ on both side of the equality condition to obtain

$$(z_{i,1} - z_{j,1})\hat{g}_1 + \cdots + (z_{i,k} - z_{j,k})\hat{g}_k = 0 \tag{21}$$

$$(z_{i,1} - z_{j,1})\hat{g}_\eta^T\hat{g}_1 + \cdots + (z_{i,k} - z_{j,k})\hat{g}_\eta^T\hat{g}_k = \hat{g}_\eta^T(0) = 0 \tag{22}$$

Using Eq. (18), since the inner product between $\hat{g}_\eta$ with another $\hat{g}_{\neq\eta}$ is always 0, only the $\langle \hat{g}_\eta, \hat{g}_\eta \rangle$ remains in the equation as

$$(z_{i,\eta} - z_{j,\eta})\langle \hat{g}_\eta, \hat{g}_\eta \rangle = 0. \tag{23}$$

From Eq. (23), we the following two possibilities:

- $\forall \eta, \; (z_{i,\eta} - z_{j,\eta}) = 0$, which implies $z_i = z_j$

- Or $\langle \hat{g}_\eta, \hat{g}_\eta \rangle = 0$

The 2nd condition of $\langle \hat{g}_\eta, \hat{g}_\eta \rangle = 0$ implies that no features belong to group $\eta$. It is important to realize here that $z_i$ came from the original features $x_i$ where $z_i = Gx_i$. Therefore, if no features belong to group $\eta$, $z_{i,\eta}$ and $z_{j,\eta}$ must both be 0, or $z_{i,\eta} = z_{j,\eta}$ for all $\eta$. Hence , we conclude that $z_i = z_j$ and consequently $T_G^+$ is injective.

**surjective**: The decoder is surjective becuase it maps $Z$ to $\text{Im}(T_G^+)$, i.e., the entire $\text{Im}(T_G^+)$ is being mapped. Since the decoder is simultaneoulsy injective and surjective, we conclude that it is also bijective. Using Lemma 3, we can conclude that mutual information is preserved where

$$I(\mathbf{Z}; \mathbf{Y}) = I(T_G^+(\mathbf{Z}); \mathbf{Y}). \tag{24}$$

$\square$

**Lemma 2.** *If $k < d$ then $T_G$ is not injective.*

*Proof.* Given the $T_G$ is a linear function, we know that if $T_G$ is injective $\iff \dim(\ker(T_G)) = 0$, where $\ker(T_G) = \{\mathbf{v} \in R^d \,|\, T_G(\mathbf{v}) = 0\}$. Moreover, for a linear transformation $T$, we have the following property:

$$\dim(\ker(T_G)) + \dim(\text{Im}(T_G)) = d$$

Because $\dim(\text{Im}(T_G)) \leq k \rightarrow \dim(\ker(T_G)) \geq d - k > 0$ Thus, $T_G$ is not injective.

$\square$

# C  Proof for Theorems 1 and 2

**Note on notation:**  We denote $H(\mathbf{X})$ as the entropy of $\mathbf{X}$.

**Theorem 1.**  *The maximum mutual information $I(\hat{\mathbf{X}}; \mathbf{X})$ is achieved if and only if its characteristic features $\mathbf{Z}$ induced by the model makes $\mathbf{X}$ representative redundant based on Def. (1), i.e.*

$$\max_G I(\hat{\mathbf{X}}; \mathbf{X}) = I(\mathbf{X}; \mathbf{X}) \iff \min_G I(\mathbf{X}; \mathbf{X}|\mathbf{Z}) = 0,$$

$$\text{s.t. } G \in \{0,1\}^{k \times d}, \sum_{i=1}^{k} G_{ij} = 1, \mathbf{Z} = T_G(\mathbf{X}), \hat{\mathbf{X}} = \psi_{\theta_G}(\mathbf{X}). \tag{25}$$

*Proof.*    We first prove the condition

$$I(\hat{\mathbf{X}}; \mathbf{X}) = I(\mathbf{X}; \mathbf{X}) \implies I(\mathbf{X}; \mathbf{X}|\mathbf{Z}) = 0. \tag{26}$$

Based on Lemma 4 in App.A, we know that $I(\hat{\mathbf{X}}; \mathbf{X})$ is upper bounded by $I(\mathbf{X}; \mathbf{X})$, i.e., the mutual information can never exceed the entropy of $\mathbf{X}$. Therefore, the optimal $G^*$ that maximizes $I(\hat{\mathbf{X}}; \mathbf{X})$ is found if the following condition is satisfied.

$$I(\hat{\mathbf{X}}; \mathbf{X}) = I(\mathbf{X}; \mathbf{X}). \tag{27}$$

This can be proven given the following three observations:

- Using Lemmas 1 and 3, MI is preserved under $T_G^+$ and therefore the information between the following are the same.
$$I(\mathbf{Z}; \mathbf{X}) = I(\hat{\mathbf{X}}; \mathbf{X}). \tag{28}$$
Using $I(\hat{\mathbf{X}}; \mathbf{X}) = I(\mathbf{X}; \mathbf{X})$. and Eq. (28) implies that
$$I(\mathbf{Z}; \mathbf{X}) = I(\mathbf{X}; \mathbf{X}). \tag{29}$$

- We note that $\mathbf{Z} = T_G(\mathbf{X})$ and that $T_G$ is a deterministic function. Therefore, $P(\mathbf{Z}|\mathbf{X}) = \delta(\mathbf{Z})$ where $\delta$ denotes the Kronecker's delta. Given this observation, we have
$$P(\mathbf{X}, \mathbf{Z}) = P(\mathbf{X}, \mathbf{Z}) \tag{30}$$
$$= P(\mathbf{Z}|\mathbf{X})P(\mathbf{X}) \tag{31}$$
$$= [\delta(T_G(\mathbf{X}))]P(\mathbf{X}) \tag{32}$$
$$P(\mathbf{X}, \mathbf{Z}) = P(\mathbf{X}). \tag{33}$$
Therefore, it leads to the conclusion that
$$I(\mathbf{X}, \mathbf{Z}; \mathbf{X}) = I(\mathbf{X}; \mathbf{X}). \tag{34}$$

- We write the chain rule for $I(\mathbf{X}, \mathbf{Z}; \mathbf{X})$ as
$$I(\mathbf{X}, \mathbf{Z}; \mathbf{X}) = I(\mathbf{Z}; \mathbf{X}) + I(\mathbf{X}; \mathbf{X}|\mathbf{Z}). \tag{35}$$
By using Eq. (34) and (29), the equality becomes
$$I(\mathbf{X}; \mathbf{X}) = I(\mathbf{X}; \mathbf{X}) + I(\mathbf{X}; \mathbf{X}|\mathbf{Z}). \tag{36}$$
Therefore, $I(\mathbf{X}; \mathbf{X}|\mathbf{Z})$ must equal to 0 and condition Eq. (26) is proven.

We next prove the reverse condition

$$I(\mathbf{X}; \mathbf{X}|\mathbf{Z}) = 0 \implies I(\hat{\mathbf{X}}; \mathbf{X}) = I(\mathbf{X}; \mathbf{X}). \tag{37}$$

Start by leveraging the result from Eq. (34) to obtain

$$I(\mathbf{X}; \mathbf{X}) = I(\mathbf{X}, \mathbf{Z}; \mathbf{X}) \tag{38}$$
$$= I(\mathbf{Z}; \mathbf{X}) + I(\mathbf{X}; \mathbf{X}|\mathbf{Z}) \qquad \text{using the chain rule} \tag{39}$$
$$= I(\mathbf{Z}; \mathbf{X}) + 0 \qquad \text{using the condition assumption} \tag{40}$$
$$= I(\hat{\mathbf{X}}; \mathbf{X}) \qquad \text{using the Eq. (28)} \tag{41}$$
$$= H(\mathbf{X}) \qquad \text{using the definition of Entropy.} \tag{42}$$

$\square$

**Theorem 2.** *The maximum mutual information $I(\bar{\mathbf{X}}; \mathbf{X})$ is achieved if and only if its $m$-selected characteristic features $\mathbf{Z} \odot \mathbf{s}$ induced by the model makes $\mathbf{X}$ relevant redundant based on Def. (2), i.e.*

$$\max_G I(\bar{\mathbf{X}}; \mathbf{Y}) = I(\mathbf{X}; \mathbf{Y}) \iff \min_G I(\mathbf{X}; \mathbf{Y}|\mathbf{Z} \odot \mathbf{s}) = 0,$$

$$\text{s.t. } G \in \{0,1\}^{k \times d}, \sum_{i=1}^{k} G_{ij} = 1, \mathbf{Z} = T_G(\mathbf{X}), \ \bar{\mathbf{X}} = \phi_{\theta_S, \theta_G}(\mathbf{X}), \mathbf{s} \in \{0,1\}^k, |\mathbf{s}| = m. \tag{43}$$

*Proof.* **Forward direction.**

The proof follows the similar direction as Theorem 1: First we show:

$$I(\bar{\mathbf{X}}; \mathbf{Y}) = I(\mathbf{X}; \mathbf{Y}) \implies I(\mathbf{X}; \mathbf{Y}|\mathbf{Z} \odot \mathbf{s}) = 0.$$

using the data processing lemma 4, $I(\bar{\mathbf{X}}; \mathbf{Y}) \leq I(\mathbf{X}; \mathbf{Y})$, hence the optimal solution $G^*, \mathbf{s}^*$ satisfies $I(\bar{\mathbf{X}}; \mathbf{Y}) = I(\mathbf{X}; \mathbf{Y})$. The goal is to show that given

$$I(\bar{\mathbf{X}}; \mathbf{Y}) = I(\mathbf{X}; \mathbf{Y}) \tag{44}$$

then

$$I(\mathbf{X}; \mathbf{Y}|\mathbf{Z} \odot \mathbf{s}) = 0. \tag{45}$$

As stated in Lemma 1 the mutual information is preserved under the decoder map, hence

$$I(\mathbf{Z} \odot \mathbf{s}; \mathbf{Y}) = I(T_G^+(\mathbf{Z} \odot \mathbf{s}); \mathbf{Y}) = I(\bar{\mathbf{X}}; \mathbf{Y}) = I(\mathbf{X}; \mathbf{Y}). \tag{46}$$

From this observation, it leads to the following derivation:

$$I(\mathbf{X}, \mathbf{Z} \odot \mathbf{s}; \mathbf{Y}) = I(\mathbf{Z} \odot \mathbf{s}; \mathbf{Y}) + I(\mathbf{X}; \mathbf{Y}|\mathbf{Z} \odot \mathbf{s}) \quad \text{via chain rule} \tag{47}$$

$$I(\mathbf{X}; \mathbf{Y}) = I(\mathbf{Z} \odot \mathbf{s}; \mathbf{Y}) + I(\mathbf{X}; \mathbf{Y}|\mathbf{Z} \odot \mathbf{s}) \quad \text{via Eq. (33)}, P(\mathbf{X}, \mathbf{Z}) = P(\mathbf{X}) \tag{48}$$

$$I(\mathbf{X}; \mathbf{Y}) = I(\mathbf{X}; \mathbf{Y}) + I(\mathbf{X}; \mathbf{Y}|\mathbf{Z} \odot \mathbf{s}) \quad \text{via Eq. (46)}, I(\mathbf{Z} \odot \mathbf{s}; \mathbf{Y}) = I(\mathbf{X}; \mathbf{Y}). \tag{49}$$

Since $I(\mathbf{X}; \mathbf{Y}) = I(\mathbf{X}; \mathbf{Y})$, then the condition $I(\mathbf{X}; \mathbf{Y}|\mathbf{Z} \odot \mathbf{s}) = 0$ must be true.

**Reverse direction.** We now prove

$$I(\mathbf{X}; \mathbf{Y}|\mathbf{Z} \odot \mathbf{s}) = 0 \implies I(\bar{\mathbf{X}}; \mathbf{Y}) = I(\mathbf{X}; \mathbf{Y}). \tag{50}$$

First note that

$$I(\mathbf{X}; \mathbf{Y}) \leq I(\mathbf{X}; \mathbf{Y}) + I(\mathbf{Z} \odot \mathbf{s}; \mathbf{Y}|X) \tag{51}$$

$$\leq I(\mathbf{X}, \mathbf{Z} \odot \mathbf{s}; \mathbf{Y}) \qquad\qquad \text{Chain Rule.} \tag{52}$$

Second, note that $\phi_{\theta_S, \theta_G}$ is a deterministic function, and therefore a function $f$ defined as $f(\mathbf{X}) = [\mathbf{X}, \phi_{\theta_S, \theta_G}(\mathbf{X})]^T$ is also a deterministic function. This give us the following relationship

$$I(\mathbf{X}, \mathbf{Y}) \geq I(f(\mathbf{X}), \mathbf{Y}) \qquad\qquad \text{Apply Lemma. 4.} \tag{53}$$

$$\geq I(\mathbf{X}, \phi_{\theta_S, \theta_G}(\mathbf{X}); \mathbf{Y}) \qquad\qquad \text{Apply function } f. \tag{54}$$

$$\geq I(\mathbf{X}, \mathbf{Z} \odot \mathbf{s}; \mathbf{Y}) \qquad\qquad \text{Apply definition of } \phi_{\theta_S, \theta_G}(X). \tag{55}$$

Combining Eq. (52) and Eq. (55) together, we have

$$I(\mathbf{X}, \mathbf{Z} \odot \mathbf{s}; \mathbf{Y}) \geq I(\mathbf{X}, \mathbf{Y}) \geq I(\mathbf{X}, \mathbf{Z} \odot \mathbf{s}; \mathbf{Y}), \tag{56}$$

which is only possible if

$$I(\mathbf{X}, \mathbf{Z} \odot \mathbf{s}; \mathbf{Y}) = I(\mathbf{X}, \mathbf{Y}). \tag{57}$$

Leveraging this result, we see that

$$I(\mathbf{X}; \mathbf{Y}) = I(\mathbf{X}, \mathbf{Z} \odot \mathbf{s}; \mathbf{Y}) \tag{58}$$

$$= I(\mathbf{Z} \odot \mathbf{s}; \mathbf{Y}) + I(\mathbf{X}; \mathbf{Y}|\mathbf{Z} \odot \mathbf{s}) \qquad\qquad \text{using the chain rule} \tag{59}$$

$$= I(\mathbf{Z} \odot \mathbf{s}; \mathbf{Y}) + 0 \qquad\qquad \text{using the condition assumption} \tag{60}$$

$$= I(\bar{\mathbf{X}}; \mathbf{Y}) \qquad\qquad \text{using the Eq. (29)} \tag{61}$$

$$\square$$

# D  Proof for Theorem 3

**Theorem 3.** *Let $\mathcal{X} = \{X_1, \ldots, X_d\}$ be a random variable that is consists of all features , let relevant features $\mathcal{U}$ be*

$$\mathcal{U} = \{X_j | \ I(X_j; \mathbf{Y}) \neq 0 \vee \exists \mathcal{A} \subseteq \mathcal{X} \ I(X_j; \mathbf{Y} | \mathcal{A}) \neq 0\}, \tag{62}$$

*and let irrelevant features be $\mathcal{U}^c$, then, $\exists G \in \mathcal{G}$ such that $I(T_G(\mathbf{X}); \mathbf{Y}) = I(\mathbf{X}; \mathbf{Y})$ if*

$$k \geq |\mathcal{U}| + \mathbb{1}(|\mathcal{U}| \neq d) \tag{63}$$

*where $\mathbb{1}(|\mathcal{U}| \neq d)$ is an indicator function equal to one when $|\mathcal{U}| \neq d$ and zero when $|\mathcal{U}| = d$.*

*Proof.* Set $\mathcal{U}$ is the collection of features with the property $I(X_j; \mathbf{Y}) \neq 0$ or $I(X_j; \mathbf{Y} | \mathcal{A}) \neq 0$. In other words, the first inequality implies that a features belongs to $\mathcal{U}$ if its mutual information with respect to $\mathbf{Y}$ is not 0. Yet, just because the MI between a feature and a label is 0, sometimes, their MI is no longer 0 given another set of features. Therefore, we add the 2nd condition to includes these cases. Namely, a feature belongs to $\mathcal{U}$ if it directly provide information on $\mathbf{Y}$ or if it indirectly provide information given $\mathcal{A}$.

Note that the definition of $\mathcal{U}$ is a consequences of Def. (2). Conversely, it allows us to also define its complement $\mathcal{U}^c$ as featuers that doesn't provide any information on $\mathbf{Y}$ even when it is conditioned on a set of features $\mathcal{A}$. Formally, we define $\mathcal{U}^c$ as

$$\mathcal{U}^c = \{X_j \mid I(X_j; \mathbf{Y}) = 0 \text{ and }, \forall \mathcal{A} \ I(X_j; \mathbf{Y} | \mathcal{A}) = 0\}. \tag{64}$$

To prove the theorem, we first we prove the following lemma:

**Lemma 5.** *Set $\mathcal{U}^c$ is relevant Redundant with respect to set $\mathcal{U}$, meaning:*

$$I(\mathcal{U}^c; \mathbf{Y} | \mathcal{U}) = 0$$

*Proof.* Without loss of generality assume $|\mathcal{U}^c| = n$ which we present these set of features by $\hat{x}^1, \ldots \hat{x}^n$ :

Based on chain rule, we have the following equality for $I(\hat{x}^1, \ldots \hat{x}^n, \mathcal{U}; \mathbf{Y})$:

$$I(\hat{x}^1, \ldots \hat{x}^n, \mathcal{U}; \mathbf{Y}) = I(\mathcal{U}; \mathbf{Y}) + I(\hat{x}^1; \mathbf{Y} | \mathcal{U}) + I(\hat{x}^2; \mathbf{Y} | \mathcal{U}, \hat{x}^1) + \cdots + I(\hat{x}^n; \mathbf{Y} | \mathcal{U}, \hat{x}^1, \ldots, \hat{x}^{n-1}) \tag{65}$$

Each $\hat{x}^i \in \mathcal{U}^c$, hence $I(\hat{x}^i; Y | \mathcal{A}) = 0$, which leads to the following:

$$I(\mathcal{U}^c, \mathcal{U}; \mathbf{Y}) = I(\hat{x}^1, \ldots \hat{x}^n, \mathcal{U}; \mathbf{Y}) = I(\mathcal{U}; \mathbf{Y}) + 0 + \cdots + 0. \tag{66}$$

Eq. 66 also leads to the conclusion of this lemma:

$$I(\mathcal{U}^c, \mathcal{U}; \mathbf{Y}) = I(\mathcal{U}; \mathbf{Y}) + I(\mathcal{U}^c; \mathbf{Y} | \mathcal{U}) = I(\mathcal{U}; \mathbf{Y}) \implies I(\mathcal{U}^c; \mathbf{Y} | \mathcal{U}) = 0$$

Which means $\mathcal{U}^c$ is *relevant redundant* with respect to set $\mathcal{U}$.

$\square$

**Lemma 6.** *With the definition of $\mathcal{U}$, we have the following equality:*

$$I(\mathcal{U}; \mathbf{Y}) = I(\mathcal{U}, \mathcal{U}^c; \mathbf{Y})$$

*Proof.* This lemma is one of the consequences of Lemma 5, using chain rules would lead us to the following results:

$$I(\mathbf{Y}; \mathcal{U}, \mathcal{U}^c) = I(\mathbf{Y}; \mathcal{U}) + I(\mathbf{Y}; \mathcal{U}^c | \mathcal{U}) = I(\mathbf{Y}; \mathcal{U}) + 0 \tag{67}$$

$\square$

Now we prove the following Lemma, which is the final step for proofing the theorem.

**Lemma 7.** *The encoder, $T_G : \mathbf{X} \to \mathbf{Z}$, is a mapping induced by a $G \in \{0,1\}^{k \times d}$, $\sum_{i=1}^{k} G_{ij} = 1$, . If $T_G$ has the property of*

$$T_G|_U := U \to \text{Im}(T_G|_U) \quad \text{is bijective} \quad \text{and} \quad \text{Im}(T_G|_U) \cap \text{Im}(T_G|_{U^c}) = \{0\} \qquad (68)$$

*where $\dim(U) = |\mathcal{U}|$, is subspace created by features in $\mathcal{U}$, and all the elements $\mathcal{U}^c$ maps to a separated axis' that is orthogonal to elements that are mapped from $U$, then*

$$I(\mathbf{X}; \mathbf{Y}) = I(T_G(\mathbf{X}); \mathbf{Y}). \qquad (69)$$

*Proof.* First we prove such a $G$ exists for $k = |\mathcal{U}| + \mathbb{1}(|\mathcal{U}| \neq d)$ and it can captures the mutual information between $\mathbf{X}$ and $\mathbf{Y}$. Note the encoder $T_G : \mathbf{X} \to \mathbf{Z}$ is from $\mathbb{R}^d \to \mathbb{R}^k$. let $\hat{e}^1, \ldots, \hat{e}^k$ the basis axis in $R^k$, and $\hat{x}^1, \ldots, \hat{x}^k$ the values of each axis. Assume $|\mathcal{U}| = m$, $T_G$ is bijective with respect to set $U$. Thus, without loss of generality, assume $e^1, \ldots, e^m$ axis in $U$ are mapped to $\hat{e}^1, \ldots, \hat{e}^m$ in $Z$. And axis in $U^c$ are mapped to $\hat{e}^{m+1}, \ldots, \hat{e}^k$. Hence $T_G$ acts as an identity for $U$, because $U^c$ sends to orthogonal subspace we can say the following: $I(T_G(\mathbf{X}); \mathbf{Y}) = I(U, \text{Im}(T_G|_{U^c}); \mathbf{Y})$. based on lemma 6 we have the following:

$$I(T_G(\mathbf{X}); \mathbf{Y}) = I(T_G(U), \text{Im}(T_G|_{U^c}); \mathbf{Y}) \geq I(T_G(U); \mathbf{Y}) = I(U; \mathbf{Y}) \qquad (70)$$

which $I(U; \mathbf{Y})$ we know it is equal to $I(\mathbf{X}; \mathbf{Y})$ based on lemma 6, Hence $I(T_G(\mathbf{X}); \mathbf{Y}) \geq (\mathbf{X}; \mathbf{Y})$. But where $I(T_G(\mathbf{X}); \mathbf{Y}) \leq (\mathbf{X}; \mathbf{Y})$ is based on data processing Lemma 4. Hence $I(\mathbf{X}; \mathbf{Y}) = I(T_G(\mathbf{X}); \mathbf{Y}) = I(U; \mathbf{Y})$, which concludes the proof. $\qquad \square$

Based on 7, as long as we can satisfy Eq. 68, we can guarantee the preservation of mutual information between $\mathbf{X}$ and $\mathbf{Y}$. As long as $k \geq |\mathcal{U}| + \mathbb{1}(|\mathcal{U}| \neq d)$, we can define $T_G$ to act as identity on $U$ and map $U^c$ to orthogonal subspace of $\text{Im}(T_G|_U)$ which doesn't need to be injective, i.e. it can map everything to one axis. If $|U| = d$ then $G$ is the identify map. Not that $G$ is a group matrix and has to map each input to an output, that is the reason we need at least one axis for elements in $U^c$.

$\qquad \square$

# E Proof for Theorem 4

**Theorem 4.**

*Given $\rho$ as the correlation coefficient and*

$$\mathcal{C} = \{X_j | \rho(X_j; \mathbf{Y}) \neq 0 \vee \exists \mathcal{A} \, \rho(X_j; \mathbf{Y}|\mathcal{A}) \neq 0\} \qquad (71)$$

*then: $|\mathcal{C}| \leq |\mathcal{U}|$.*

*Proof.* Let $\mathcal{C}_1 = \{X_j | \rho(X_j; \mathbf{Y}) \neq 0\}$ and $\mathcal{C}_2 = \{X_j | \exists \mathcal{A}, \rho(X_j; \mathbf{Y}|\mathcal{A}) \neq 0\}$, then

$$\mathcal{C} = \mathcal{C}_1 \cup \mathcal{C}_2. \qquad (72)$$

Also we know that $\mathcal{U} = \mathcal{U}_1 \cup \mathcal{U}_2$ if we let $\mathcal{U}_1 = \{X_j | I(X_j; \mathbf{Y}) \neq 0\}$ and $\mathcal{U}_2 = \{X_j | \exists \mathcal{A}, I(X_j; \mathbf{Y}|\mathcal{A}) \neq 0\}$. If we can show that $\mathcal{C}_1 \subseteq \mathcal{U}_1$ and $\mathcal{C}_2 \subseteq \mathcal{U}_2$, then $|\mathcal{C}| \leq |\mathcal{U}|$ is proven.

We first note that since MI of $I(Z_1; Z_2|\mathcal{A})$ measures the KL divergence between $P(Z_1, Z_2|\mathcal{A})$ and $P(Z_1|\mathcal{A})P(Z_2|\mathcal{A})$, if $I(Z_1; Z_2|\mathcal{A}) = 0$, then the following condition must also be true:

$$P(Z_1, Z_2|\mathcal{A}) = P(Z_1|\mathcal{A})P(Z_2|\mathcal{A}). \qquad (73)$$

Using the condition from Eq. (73), we can compute the conditional expectation where

$$E_{Z_1, Z_2}[Z_1 Z_2] = \int_{z_1} \int_{z_2} z_1 z_2 p(z_1, z_2|\mathcal{A}) dz_1 dz_2 \qquad (74)$$

$$= \int_{z_1} \int_{z_2} z_1 z_2 p(z_1|\mathcal{A}) p(z_2|\mathcal{A}) dz_1 dz_2 \qquad (75)$$

$$= \left[ \int_{z_1} z_1 p(z_1|\mathcal{A}) dz_1 \right] \left[ \int_{z_2} z_2 p(z_2|\mathcal{A}) dz_2 \right] \qquad (76)$$

$$= E_{Z_1}[Z_1] E_{Z_2}[Z_2]. \qquad (77)$$

Since $E_{Z_1,Z_2}[Z_1 Z_2] = E_{Z_1}[Z_1] E_{Z_2}[Z_2]$, it implies that the cross-covariance must also be 0 where

$$E_{Z_1,Z_2}[Z_1 Z_2] - E_{Z_1}[Z_1] E_{Z_2}[Z_2] = 0. \tag{78}$$

Since $\rho$ is the cross-covariance scaled by a constant, we see that if MI is 0 then $\rho$ is also 0. By contrapositivity, we see that if $\rho$ is not equal to zero then MI is not equal to 0. Therefore, if an element is in $\mathcal{C}_1$ or $\mathcal{C}_2$, it must also be included into $\mathcal{U}_1$ or $\mathcal{U}_2$. Hence, we have shown that $\mathcal{C}_1 \subseteq \mathcal{U}_1$ and $\mathcal{C}_2 \subseteq \mathcal{U}_2$

$\square$

**Corollary 4.1.** *Theorems 3 and 4 yields a lower bound for k where $|\mathcal{C}| + \mathbb{1}(|\mathcal{C}| \neq d) \leq k$.*

*Proof.* We want to proof that $|\mathcal{C}| + \mathbb{1}(|\mathcal{C}| \neq d) \leq |\mathcal{U}| + \mathbb{1}(|\mathcal{U}| \neq d)$, and then using Theorem 3, $k \geq |\mathcal{U}| + \mathbb{1}(|\mathcal{U}| \neq d)$, we conclude the proof. Using Theorem 4, we have $\mathcal{C} \subseteq \mathcal{U}$. We have the following cases:

- **Case 1.** $|\mathcal{C}| = d$: Since $\mathcal{C} \subseteq \mathcal{U}$, thus $|\mathcal{U}| = d$. and therefore: $|\mathcal{C}| + \mathbb{1}(|\mathcal{C}| \neq d) = |\mathcal{U}| + \mathbb{1}(|\mathcal{U}| \neq d) \leq k$.

- **Case 2.** $|\mathcal{C}| < d$: This means $\mathbb{1}(|\mathcal{C}| \neq d) = 1$ We have two sub cases:

  - $\mathbb{1}(|\mathcal{U}| \neq d) = 1$: using the fact that $|\mathcal{C}| \leq |\mathcal{U}|$, we conclude: $|\mathcal{C}| + \mathbb{1}(|\mathcal{C}| \neq d) \leq |\mathcal{U}| + \mathbb{1}(|\mathcal{U}| \neq d)$, since both indicator function are equal to one.
  - $\mathbb{1}(|\mathcal{U}| \neq d) = 0$: This means that $|\mathcal{U}| = d$ or the whole space, since $|\mathcal{C}| < d$, therefore, there exist atleast one element in $\mathcal{U}$ that is not in $\mathcal{C}$, thus: $|\mathcal{C}| < |\mathcal{U}|$, which means: $\mathbb{1}(|\mathcal{C}| \neq d) \leq |\mathcal{U}| + \mathbb{1}(|\mathcal{U}| \neq d)$.

$\square$

**Corollary 4.2.** For Gaussian distributions the inequality turns into equality where $|\mathcal{C}| = |\mathcal{U}|$.

*Proof.* If $\mathbf{X}$ and $\mathbf{Y}$ are independent, then they are also uncorrelated. However, if $\mathbf{X}$ and $\mathbf{Y}$ are uncorrelated, then they could be dependent. General case when lack of correlation implies independence is when the joint distribution of $\mathbf{X}$ and $\mathbf{Y}$ is Gaussian, which means $\mathcal{C} = \mathcal{U}$. Note that in classification case, $P(X, Y)$ cannot be Gaussian due to the discreteness of the classification problem.

$\square$

# F  Complete Collection of *Style Adaptation* Results

In figure 8, we see the complete collection of *Style Adaptation* Results. Our method, gI, is capable of capturing the most relevant pixels in spite of having multiple classes with variations among each class.

Figure 8: gI is capable to handling the added complexity of style variation among multiple classes.

## MNIST result for 3 groups

In Fig 9, we investigate gI model on $k = 3$ and $m = 1$ number of groups we showed each group with a different color red, blue, green. As we can see increasing the number of groups kept the shape of each digits but it separates one of the groups into two which is the digit patterns

Figure 9: MNIST result for $k = 3$ groups and $m = 1$ number of most important groups.

## MNIST alphabet

Im Fig 10, we tried our model for classification of 25 alphabet and the result of grouping is as follows for some of the instances.

Figure 10: MNIST Alphabet

**COPD result for 3 Groups.**

In Fig 11 we tried COPD gene data set with $k = 3$ number of groups and we are showing the important group with white pixels similar to the paper. The result are indicator of similar network as we have for 2 groups in the paper but increasing number of groups seems to make the important group sparser or less number of features in the most important group.

Figure 11: COPD gene expression and gene most important groups for $k = 3$, $m = 1$.

**Implementation details for experiments**.

Since an overly expressive network of $Q_{\theta_R}$ for reconstruction confounds the interpretability of the group structure. It is preferable to define $Q_{\theta_R}$ as a simpler function. We have found two functions that work well experimentally. First, for *balanced* dataset, $Q_{\theta_R}$ can simply be an identify function. However, for *unbalanced* datasets, we found a well behaved function to be one that converts each characteristic feature to the average value of features in that group. This can be seen as the following function: $f_{\theta_R} : \mathbb{R}^d \to \mathbb{R}^b$ where given $I_i$ to be the indexes of feature belong to group $i$. And let $j \in I_j$ It send $\hat{x}_j$ to $f_{\theta_R}(\hat{x}_j) = \frac{\hat{x}_j}{|I_i|}$ . which means the average value of each group is a good reconstruction of each feature in that group.

## G   Variational lower bound

From Eq. (7), Eq. (8) we can derive the graphical model which is shown on Fig. 12.

Figure 12: Graphical model for group learning representation

Given our network $\phi_{\theta_S, \theta_G}$ that is parameterized by $\theta_G, \theta_S$. We wish to solve the problem

$$\underset{\theta_G, \theta_S}{\arg\max} \quad \underbrace{I(\psi_{\theta_G}(\mathbf{X}), \mathbf{X})}_{\Xi_1} + \underbrace{\lambda I(\phi_{\theta_S, \theta_G}(\mathbf{X}); \mathbf{Y})}_{\Xi_2} . \tag{79}$$

MI, however, is difficult to compute due to its requirement of both the joint and marginal distributions. Chen et al. [11] circumvented this problem by calculating the variational lower bound. Since the lower bound is tractable, it can be maximized as a surrogate to Eq. (**??**). We start the derivation by

only looking at $\Xi_2$ following the definition of MI as

$$\underset{\theta_G,\theta_S}{\arg\max}\, \mathrm{MI}(\phi_{\theta_S,\theta_G}(\mathbf{X});\mathbf{Y}) = \underset{\theta_G,\theta_S}{\arg\max} \int_{x\in\mathcal{X}}\int_{y\in\mathcal{Y}} p(\phi_{\theta_S,\theta_G}(x),y)\mathrm{log}\left(\frac{p(\phi_{\theta_S,\theta_G}(x),y)}{p(\phi_{\theta_S,\theta_G}(x))p(y)}\right)dxdy. \tag{80}$$

Since $p(\phi_{\theta_S,\theta_G}(x),y) = p(y|\phi_{\theta_S,\theta_G}(x))p(\phi_{\theta_S,\theta_G}(x))$, we replace the numerator term inside the log and cancel out $p(\phi_{\theta_S,\theta_G}(x))$, the objective then becomes

$$\underset{\theta_G,\theta_S}{\arg\max}\, \mathrm{MI}(\phi_{\theta_S,\theta_G}(\mathbf{X});\mathbf{Y}) = \underset{\theta_G,\theta_S}{\arg\max} \int_{x\in\mathcal{X}}\int_{y\in\mathcal{Y}} p(\phi_{\theta_S,\theta_G}(x),y)\mathrm{log}\left(\frac{p(y|\phi_{\theta_S,\theta_G}(x))}{p(y)}\right)dxdy. \tag{81}$$

The integrals can be rewritten into

$$= \underset{\theta_G,\theta_S}{\arg\max} \int_{x\in\mathcal{X}}\int_{y\in\mathcal{Y}} p(\phi_{\theta_S,\theta_G}(x),y)\mathrm{log}\left[p(y|\phi_{\theta_S,\theta_G}(x))\right] - p(\phi_{\theta_S,\theta_G}(x),y)\mathrm{log}\left[p(y)\right]dxdy,$$

$$= \underset{\theta_G,\theta_S}{\arg\max} \int_{x\in\mathcal{X}}\int_{y\in\mathcal{Y}} p(\phi_{\theta_S,\theta_G}(x),y)\mathrm{log}\left[p(y|\phi_{\theta_S,\theta_G}(x))\right]dxdy - \int_{x\in\mathcal{X}}\int_{y\in\mathcal{Y}} p(\phi_{\theta_S,\theta_G}(x),y)\mathrm{log}\left[p(y)\right]dxdy,$$

$$= \underset{\theta_G,\theta_S}{\arg\max} \int_{x\in\mathcal{X}}\int_{y\in\mathcal{Y}} p(\phi_{\theta_S,\theta_G}(x),y)\mathrm{log}\left[p(y|\phi_{\theta_S,\theta_G}(x))\right]dxdy - \int_{y\in\mathcal{Y}}\left[\int_{x\in\mathcal{X}} p(\phi_{\theta_S,\theta_G}(x),y)dx\right]\mathrm{log}\left[p(y)\right]dy,$$

$$= \underset{\theta_G,\theta_S}{\arg\max} \int_{x\in\mathcal{X}}\int_{y\in\mathcal{Y}} p(\phi_{\theta_S,\theta_G}(x),y)\mathrm{log}\left[p(y|\phi_{\theta_S,\theta_G}(x))\right]dxdy - \int_{y\in\mathcal{Y}} p(y)\mathrm{log}\left[p(y)\right]dy.$$

Since $\int_{y\in\mathcal{Y}} p(y)\mathrm{log}\left[p(y)\right]dy$ no longer has a $\theta_G,\theta_S$ term, the maximization over $\theta_G,\theta_S$ can be treated as a constant, i.e., this term can be removed from the optimization object which leads us to

$$\underset{\theta_G,\theta_S}{\arg\max} \int_{x\in\mathcal{X}}\int_{y\in\mathcal{Y}} p(\phi_{\theta_S,\theta_G}(x),y)\mathrm{log}\left[p(y|\phi_{\theta_S,\theta_G}(x))\right]dxdy, \tag{82}$$

$$\underset{\theta_G,\theta_S}{\arg\max} \int_{x\in\mathcal{X}}\int_{y\in\mathcal{Y}} p(y|\phi_{\theta_S,\theta_G}(x))p(\phi_{\theta_S,\theta_G}(x))\mathrm{log}\left[p(y|\phi_{\theta_S,\theta_G}(x))\right]dxdy, \tag{83}$$

$$\underset{\theta_G,\theta_S}{\arg\max} \int_{x\in\mathcal{X}} p(\phi_{\theta_S,\theta_G}(x))\left[\int_{y\in\mathcal{Y}} p(y|\phi_{\theta_S,\theta_G}(x))\mathrm{log}\left[p(y|\phi_{\theta_S,\theta_G}(x))\right]dy\right]dx. \tag{84}$$

$$\underset{\theta_G,\theta_S}{\arg\max}\, E_{\phi_{\theta_S,\theta_G}(\mathbf{X})}\left[\int_{y\in\mathcal{Y}} p(y|\phi_{\theta_S,\theta_G}(x))\mathrm{log}\left[p(y|\phi_{\theta_S,\theta_G}(x))\right]dy\right], \tag{85}$$

$$\underset{\theta_G,\theta_S}{\arg\max}\, E_{\phi_{\theta_S,\theta_G}(\mathbf{X})}E_{\mathbf{Y}|\phi_{\theta_S,\theta_G}(\mathbf{X})}\left[\mathrm{log}(p(\mathbf{Y}|\phi_{\theta_S,\theta_G}(\mathbf{X})))\right] =$$
$$\underset{\theta_G,\theta_S}{\arg\max}\, E_{\mathbf{Y},\phi_{\theta_S,\theta_G}(\mathbf{X})}\left[\mathrm{log}(p(\mathbf{Y}|\phi_{\theta_S,\theta_G}(\mathbf{X})))\right]. \tag{86}$$

If we look closer at the inner expectation, $E_{\mathbf{Y}|\phi_{\theta_S,\theta_G}(\mathbf{X})}\left[log(p(\mathbf{Y}|\phi_{\theta_S,\theta_G}(\mathbf{X})))\right]$, we do not assume to have $p(\mathbf{Y}|\phi_{\theta_S,\theta_G}(\mathbf{X}))$. Instead, we wish to approximate the distribution via another distribution $Q_{\theta_P}(\mathbf{Y}|\phi_{\theta_S,\theta_G}(\mathbf{X}))$ that is parameterized by $\theta_P$. The approximation can be done by making sure that the KL divergence between $p$ and $q$ is minimized. When writing out the KL divergence, we get

$$KL(p||q) = \int_{y\in\mathcal{Y}} p(y|\phi_{\theta_S,\theta_G}(x))\mathrm{log}\left[\frac{p(y|\phi_{\theta_S,\theta_G}(x))}{Q_{\theta_P}(y|\phi_{\theta_S,\theta_G}(x))}\right]dy,$$

$$= \int_{y\in\mathcal{Y}} p(y|\phi_{\theta_S,\theta_G}(x))\mathrm{log}(p(y|\phi_{\theta_S,\theta_G}(x))) - p(y|\phi_{\theta_S,\theta_G}(x))\mathrm{log}(Q_{\theta_P}(y|\phi_{\theta_S,\theta_G}(x)))dy,$$

$$= E_{\mathbf{Y}|\phi_{\theta_S,\theta_G}(\mathbf{X})}[\mathrm{log}p(y|\phi_{\theta_S,\theta_G}(x)))] - E_{\mathbf{Y}|\phi_{\theta_S,\theta_G}(\mathbf{X})}[\mathrm{log}Q_{\theta_P}(y|\phi_{\theta_S,\theta_G}(x)))].$$

Since KL divergence is always 0 or greater, we get the inequality relation

$$E_{\mathbf{Y}|\phi_{\theta_S,\theta_G}(\mathbf{X})}[\mathrm{log}p(y|\phi_{\theta_S,\theta_G}(x)))] - E_{\mathbf{Y}|\phi_{\theta_S,\theta_G}(\mathbf{X})}[\mathrm{log}Q_{\theta_P}(y|\phi_{\theta_S,\theta_G}(x)))] \geq 0 \tag{87}$$

$$E_{\mathbf{Y}|\phi_{\theta_S,\theta_G}(\mathbf{X})}[\mathrm{log}p(y|\phi_{\theta_S,\theta_G}(x)))] \geq E_{\mathbf{Y}|\phi_{\theta_S,\theta_G}(\mathbf{X})}[\mathrm{log}Q_{\theta_P}(y|\phi_{\theta_S,\theta_G}(x)))]. \tag{88}$$

The inequality suggests that $E_{\mathbf{Y}|\phi_{\theta_S,\theta_G}(\mathbf{X})}[\log(Q_{\theta_P}(\mathbf{Y}|\phi_{\theta_S,\theta_G}(\mathbf{X})))]$ is a lower bound of $E_{\mathbf{Y}|\phi_{\theta_S,\theta_G}(\mathbf{X})}[\log(p(\mathbf{Y}|\phi_{\theta_S,\theta_G}(\mathbf{X})))]$, and they are equal only when $p = q$. Therefore, by finding the $\theta$ that maximizes $E_{\mathbf{Y}|\phi_{\theta_S,\theta_G}(\mathbf{X})}[\log(Q_{\theta_P}(\mathbf{Y}|\phi_{\theta_S,\theta_G}(\mathbf{X})))]$ is equivalent to finding the best approximation of $E_{\mathbf{Y}|\phi_{\theta_S,\theta_G}(\mathbf{X})}[\log(p(\mathbf{Y}|\phi_{\theta_S,\theta_G}(\mathbf{X})))]$. Next, we take Inequality (88) and rewrite each term back in terms of its integration, we obtain

$$
\int_{y\in\mathcal{Y}} p(y|\phi_{\theta_S,\theta_G}(x))\log p(y|\phi_{\theta_S,\theta_G}(x)))dy \geq
$$
$$
\int_{y\in\mathcal{Y}} p(y|\phi_{\theta_S,\theta_G}(x))\log Q_{\theta_P}(y|\phi_{\theta_S,\theta_G}(x)))dy. \tag{89}
$$

The key realization of this inequality is that given any $\phi_{\theta_S,\theta_G}(\mathbf{X})$, the inequality will still hold. Therefore, if we additionally integrate both terms over any set of $\mathcal{X}$, the inequality will still hold. Following this logic, we can add an additional integration and maintain the inequality.

$$
\int_{x\in\mathcal{X}} p(\phi_{\theta_S,\theta_G}(x)) \int_{y\in\mathcal{Y}} p(y|\phi_{\theta_S,\theta_G}(x))\log p(y|\phi_{\theta_S,\theta_G}(x)))dydx \geq
$$
$$
\int_{x\in\mathcal{X}} p(\phi_{\theta_S,\theta_G}(x)) \int_{y\in\mathcal{Y}} p(y|\phi_{\theta_S,\theta_G}(x))\log Q_{\theta_P}(y|\phi_{\theta_S,\theta_G}(x)))dydx \tag{90}
$$

$$
\int_{x\in\mathcal{X}} \int_{y\in\mathcal{Y}} p(y,\phi_{\theta_S,\theta_G}(x))\log p(y|\phi_{\theta_S,\theta_G}(x)))dydx \geq
$$
$$
\int_{x\in\mathcal{X}} \int_{y\in\mathcal{Y}} p(y,\phi_{\theta_S,\theta_G}(x))\log Q_{\theta_P}(y|\phi_{\theta_S,\theta_G}(x)))dydx. \tag{91}
$$

$$
E_{\mathbf{Y},\phi_{\theta_S,\theta_G}(\mathbf{X})}\left[\log(p(\mathbf{Y}|\phi_{\theta_S,\theta_G}(\mathbf{X})))\right] \geq E_{\mathbf{Y},\phi_{\theta_S,\theta_G}(\mathbf{X})}\left[\log(Q_{\theta_P}(\mathbf{Y}|\phi_{\theta_S,\theta_G}(\mathbf{X})))\right] \tag{92}
$$

By looking at the relationship between Eq. (86) and (92), notice that if we simultaneously maximize $\theta$ and $\theta_G, \theta_S$ using $Q_{\theta_P}$, the $\theta_P$ term would help us find the closest approximation of $p$ while the $\theta_G, \theta_S$ term would help us maximize the MI objective. Therefore, to maximize Eq. (??), we can use $Q_{\theta_P}$ as a surrogate and instead maximize

$$
\max_{\theta_P,\theta_G,\theta_S} E_{\mathbf{Y},\phi_{\theta_S,\theta_G}(\mathbf{X})}\left[\log(Q_{\theta_P}(\mathbf{Y}|\phi_{\theta_S,\theta_G}(\mathbf{X})))\right] \tag{93}
$$

We can estimate Eq. 93 by ancestral sampling from $\mathbf{X}$ and $\mathbf{Y}$ based on the graphical model in Fig. 12 to compute the expectation empirically in the new objective below as

$$
\max_{\theta_P,\theta_G,\theta_S} \frac{1}{n}\sum_{i=1} \log(Q_{\theta_P}(y_i|\phi_{\theta_S,\theta_G}(x_i))). \tag{94}
$$

Here, we are performing maximum likelihood. Note that since $\mathbf{Y}$ is the label, the probability of $y_i$ equaling its label is 1, and the probability of it being another label is 0. Therefore, the Eq. (94) can be equivalently written as minimizing the Cross-Entropy loss where

$$
\min_{\theta_P,\theta_G,\theta_S} -\sum_{i=1} p(y_i)\log(Q_{\theta_P}(y_i|\phi_{\theta_S,\theta_G}(x_i))). \tag{95}
$$

Therefore, given $\psi_{\theta_G,\theta_S}(\mathbf{X}) = \bar{\mathbf{X}}$, objective (95) can be used in place of the $\Xi_2$ term of our objective Eq. (79).

Following the same derivation, we can replace the $\Xi_1$ with its lower bound as well. Here we use $Q_{\theta_R}$ as the neural network to approximate the true distribution.

$$
\max_{\theta_R,\theta_G} E_{\mathbf{X},\psi_{\theta_G}(\mathbf{X})}\left[\log(Q_{\theta_R}(\mathbf{X}|\psi_{\theta_G}(\mathbf{X})))\right]. \tag{96}
$$

Consequently, instead of maximizing Eq. (79) directly, we can maximized its variational lower bound

$$
\max_{\theta_R,\theta_G,\theta_S} E_{\mathbf{X},\psi_{\theta_G}(\mathbf{X})}\left[\log(Q_{\theta_R}(\mathbf{X}|\psi_{\theta_G}(\mathbf{X})))\right] + \lambda E_{\mathbf{Y},\phi_{\theta_S,\theta_G}(\mathbf{X})}\left[\log(Q_{\theta_P}(\mathbf{Y}|\phi_{\theta_S,\theta_G}(\mathbf{X})))\right], \tag{97}
$$

or

$$
\max_{\theta_P,\theta_R,\theta_G,\theta_S} E_{\mathbf{X},\hat{\mathbf{X}}}\left[\log(Q_{\theta_R}(\mathbf{X}|\hat{\mathbf{X}}))\right] + \lambda E_{\mathbf{Y},\bar{\mathbf{X}}}\left[\log(Q_{\theta_P}(\mathbf{Y}|\bar{\mathbf{X}}))\right]. \tag{98}
$$

**Different Variations of this Objective.** It is not always necessary to approximate $p(\mathbf{X}|\hat{\mathbf{X}})$ with $Q_{\theta_R}$. Depending on prior information on the data, we can simply assume $p(\mathbf{X}|\hat{\mathbf{X}})$ to have a certain distribution. For example, we can simply set it to the Gaussian distribution if $\mathbf{X}$ can take any value. Here, if we let $Q_{\theta_P}$ be a Gaussian distribution of some constant $\sigma$, then given $\psi_{\theta_G}(x_i)$ as the mean Eq. (96) becomes

$$\max_{\theta_G, \theta_S} \frac{1}{n} \sum_{i=1} \log \left( e^{-\frac{||x_i - \psi_{\theta_G}(x_i)||^2}{2\sigma^2}} \right). \tag{99}$$

By building $\sigma$ directly into $\lambda$, we can ignore $\sigma$. By applying the log term to the exponential term, the objective becomes

$$\min_{\theta_G, \theta_S} \sum_{i=1} ||x_i - \psi_{\theta_G, \theta_S}(x_i)||^2. \tag{100}$$

From this objective, we see that by assuming that $Q_{\theta_R}$ is a Gaussian Distribution, we can instead optimize MSE as a variational lower bound for the mutual information, i.e., we no longer need to pass $\psi_{\theta_G}(x_i)$ through $Q_{\theta_R}$. Therefore, as a surrogate, we can solve Eq. (3) with

$$\min_{\theta_P, \theta_R, \theta_G, \theta_S} \sum_{i=1} ||x_i - \psi_{\theta_G, \theta_S}(x_i)||^2 - \lambda \sum_{i=1} p(x_i) \log(Q_{\theta_P}(x_i | \phi_{\theta_S, \theta_G}(x_i))). \tag{101}$$

# H Computational and Memory Complexity Analysis

We derive the complexity for a general $k, d$, but in most cases, we assume the number of group are much smaller than number of features meaning: $k << d$. For our purposes, we are using stochastic gradient descent. So the complexity is proportion to the number of samples $N$ into the number of parameters involved in the neural net. For each sample, the input size is $d$, then we have an neural net which the output is the Group matrix $G$, hence the number of parameters for this part is $kd^2$. Since $G$ determines the auto-encoder $\psi_{\theta_G}(x_i) = G^T G x_i$ for a sample $x_i$. thus that is just matrix multiplication. So for auto-encoder $\psi_{\theta_G}(x_i)$ the complexity is $O(nkd^2)$. For group selection part, we have another neural net for learning the projection map through selector $\mathbf{s}$, which the complexity is $k^2$ which lead to $\mathbf{Z} \odot \mathbf{s}$. Having $\mathbf{Z} \odot \mathbf{s}$, lead to $\bar{\mathbf{X}}$ by matrix multiplication. We used the last neural net from $\bar{\mathbf{X}}$ to predict the class lables. Assuming we have $C$ classes, lead to the complexity of $Cd$. Hence the overall complexity for $\mathbf{gI}$ is $O(n(kd^2 + k^2 + Cd))$, assuming $C << d$ and $k << d$, complexity of $\mathbf{gI}$ is $O(nkd^2)$. For the memory complexity of a stochastic gradient descent, we only need to save the information of weights for each sample at a time, hence it is $O(kd^2)$. If we are doing mini-batch this number linearly increases by the size of the mini-batches.