[Reviews · NeurIPS 2020]

Review 1

Summary and Contributions: I have read the authors rebuttal and others reviews. I maintain my score. ______ This article proposes a variable selection procedure that takes into account both the labels and the interactions with other features. To do so, the authors formally define notions of "redundancy" by means of both mutual information (MI) within set of features (to measure representativeness of a given variable) and MI wrt labels (that measures prediction impact). The group of feature are later selected by maximizing a (balanced) sum of this two quantities.

Strengths: The methods are rigorously presented with clear and formal definitions. The experimental evaluations show that the proposed Group Interpreter algorithm is competitive. However it is not uniformly better than old style feature selection algorithm like Lasso (see Table 2) The overall contribution is solid and transparent.

Weaknesses: The overall architecture is quite complex and leads to non convex optimization problem for approximating the exact formulation in Eq 3. This introduces computational complexity, but also lack of guarantee in the solutions obtained in practice (local minima etc ...) The paper does not present any statistical guarantee.

Correctness: I did not check all the proof's details.

Clarity: The paper is clearly written. The architecture presented in section 2 is quite complex and the description lacks intuition. For instance the choice of auto-encoder and the discussions here is quite hard to follow.

Relation to Prior Work: Fairly discussed with various comparisons.

Reproducibility: Yes

Additional Feedback: It should be interesting to display the computational overhead of the methods being compared. In some settings, they achieve fairly close performances.


Review 2

Summary and Contributions: This paper describes a framework to discover groups of related features and identify relevant groups for predicting labels on a per-sample basis. The goal is to have an interpretable latent factor model that can be used to better understand datasets with naturally modular structure, like gene expression. The technical approach is based on a customized autoencoder style architecture optimized with an information-theoretic objective. Experiments on real and synthetic data and comparisons with several comparable methods validate the approach.

Strengths: The theoretical grounding was clear and easy to follow. Empirically evaluating unsupervised learning when the goal is to find interpretable structure is difficult, but I thought the paper did a good job of including a mix of synthetic and real-world experiments, and reasonable baselines for comparison. Discovering interpretable information in an unsupervised way, especially in biomedical data like gene expression, is an important problem and this seems like a good contribution in that direction. The approach is general enough to be of wide interest to the NeurIPS community.

Weaknesses: - Validating the approach on gene expression I really wanted to see that the feature grouping per instance provided something novel for gene expression. That is, I would like to see that for smokers, we get genes A,B,C,D are all correlated (in a group for these instances), but for non-smokers, A,B,C,D are uncorrelated (not grouped for these instances). I don't think that's whats shown in Fig. 5,6, where we see that the expression levels for A,B,C,D (top rows) are low for smokers and high for non-smokers. Plots like Fig. 5,6 can be made with probably any of the global feature clustering methods. Also, there was no attempt to interpret the resulting clusters, perhaps by comparing enrichment of terms from the Gene Ontology database (though it is tricky for NeurIPS, where people won't be familiar with the domain, it still should be possible to define some sort of sensible validation of the groups besides the visual plot which is not very strong justification). - A limitation of the approach The construction of the groups relies on essentially unweighted averaging of features in a group, so the features should ideally by positively correlated, with similar correlations. In gene expression, however, you often see a mix of correlated and anti-correlated genes that should arguably be in the same group.

Correctness: Yes, the theoretical development and empirical approach were appropriate and correct.

Clarity: The paper was well presented and easy to read. The authors' version of the new "broader impacts" section was the most eloquent I've read so far.

Relation to Prior Work: Yes.

Reproducibility: Yes

Additional Feedback: A few other minor comments: Other methods to compare? For gene expression experiments, I've found k-means clustering using R^2 on columns to be a surprisingly good baseline for global feature clustering (and it allows for getting groups that are anti-correlated). I've also seen methods for finding groups of variables with high multivariate mutual information / total correlation / redundancy. Those ideas could be interesting for comparison or for improving the representation redundancy part to allow for groups that are not always positively correlated. Attention-based learning methods are natural comparisons for things like Fig. 7. Those methods really only capture the "relevant redundancy", but it might be nice to show explicitly examples where they fail to distinguish two distinct groups of relevant features. (or maybe they would tend to ignore some correlated features that are grouped together in your approach). - Isn't there a more direct ref for the Gumbel Softmax trick? - Sample complexity Can you say much about the sample complexity? The grouping mechanism is probably quite sample efficient, as it gives a strong prior. - Other real-world applications Doing MNIST and FMNIST probably makes sense as a second real-world applications as these are familiar for NeurIPS, but I think other domains might have been more appropriate. It would be fun to look at personality survey data. There, psychologists assume that questions/features are always globally correlated, but you might find that there are people who interpret the questions differently somehow. Neuroimaging also has modular structure - though it might be too high-d for your SVD step. Text documents might also be interesting. Perhaps you could disambiguate words. Finally, if you are looking for more large-scale empirical tests, openML has a lot of gene expression data. EDIT: I have read the author feedback. Even though you compare 9 methods, I think that the evaluation of *only* test accuracy is not sufficient. Since the method purports to find clusters of genes that are relevant for prediction, there should be some way to measure to what extent those clusters reflect meaningful structure. Fig. 7 gives a qualitative view of this for MNIST and Table 3 measure for the synthetic case, but I think the work would have more impact if it could demonstrate the value of the learned structure in a challenging setting like gene expression. That's also what I was looking for in Fig. 5, which I realize that I misinterpreted. Scrutinizing it now, I can see how you find clusters of genes, for each *individual*, which are relevant. Your sorting identifies the largest cluster relevant for smoking, non-smoking respectively. But again, I don't see any way of knowing whether these groups are meaningful or useful. There are a lot of "feature importance" methods applied to neural networks that produce importance of feature per sample like you do, but often the identified features are not particularly meaningful (e.g. discussed here: https://www.aaai.org/ojs/index.php/AAAI/article/view/4252).


Review 3

Summary and Contributions: This paper considers the problem of group feature extraction. The analysis is based on the group mutual information (gI). By using gI, the feature redundancy and relevant redundancies are defined. The formulation of so called "Instance-wise Feature Grouping" is thus constructed to minimize the redundancy. One main challenge of using mutual information is the difficulty of optimization, and the authors used the variational lower bound to approximate it. Another contribution is using the Gumbel-softmax layer for the index sampling. Results on both synthetic datasets and real gene datasets are provided to show the performance of the proposed model and algorithm.

Strengths: The main contribution to me is the problem formulation based on group mutual information, i.e., definitions 1 and 2 and Theorems 1 and 2. +) Optimization using the variational lower bound [11] +)Gumbel-soft-max to enable sampling +) Optimize the number of groups. +) Real data results on gene data

Weaknesses: I like the framework based on mutual information. The formulation is solid but actually not novel. -) Theorem 3 is weak and might not be useful -) The results are somehow confused. More analysis need to be given on why Lasso sometimes performs better -) Instead of plots in Figure 3, 5 and 6, some analysis on the groups of genes might be helpful. The information of these plots are low. -) Concern about the proof of theorem 1 & 2. I am confused starting from Eq. 36 in appendix. As claimed in your proof, the mutual information I(Xhat,X) is upper bounded by the entropy of X, i.e., I(x,x) = H(x). The deviation starting from Eq. 36 is plausible iif the upper bound can be reached. My main concern is that whether the upper bound is actually reachable? Since the characteristic features are the compressed version of the original features and the reconstruction will also include inversible operations (e.g., ReLU), I am a bit worried about the correctness of the proof and I am afraid whether these two theorems actually contribute to the method developing. More explanation is appreciated. -) I feel a bit confused about the MI lower bound estimation. The authors first say the samples can be generated by using Eq. 7&8 while the relationship with (P(X|Xhat), P(Y|Xbar)) is not clearly stated. However, the author also says the two distribution are unknown (P(X|Xhat), P(Y|Xbar)) and thus use another method. As such, why do you include Eq. 7&8 and the corresponding explanation.

Correctness: Most of the claims and method are correct and solid. The empirical methodology is correct but some improvement is necessary on the gene data analysis

Clarity: Yes. The paper is easy to follow.

Relation to Prior Work: The paper is based on mutual information. Probably some of the literature review on MI is needed.

Reproducibility: Yes

Additional Feedback: 1) Literature review on mutual information for feature extraction 2) More analysis on gene data 3) States clearly the difference of this work with previous group feature extraction. 4) More details in the proof --- Post Rebuttal: I have read the rebuttal and it has addressed some of my concerns on the theorem and mutual information. Therefore, I changed my score.


Review 4

Summary and Contributions: Propose an information theoretical based method to perform instance-wise feature selection, which essentially to select features that are relevant/aligned with labels as well as not been redundant.

Strengths: * The contribution appears novel and significant towards the area of feature selection and explanable AI. * The proposed method is well motivated and well designed. Authors also propose bounds and optimising on those to make the model fitting easier and faster. * Experiments combine both synthetic and real world data sets. Synthetic datasets allows some experiments that can control the amount of redundancy. The real datasets of gene and MNIST demonstrates the application of the proposed work.

Weaknesses: * I may have missed it and although dated, would be good to mention and compare with "older" measures like MRMR? See https://www.computer.org/csdl/trans/tp/2005/08/i1226-abs.html . * How to stop each instance obtaining it group of features and overwhelm users in terms of cognitive attention. * The model itself is relatively complex, it would be good if there was some ablation type of testing to show all components are necessary

Correctness: The method and proofs appears correct, although I didn't check too thoroughly. Empirical methodology is reasonable.

Clarity: The paper is generally well written. The main paper is self-contained and one can follow the arguments and definitions, lemmas etc.

Relation to Prior Work: See comment about weaknesses, but generally it seems to discuss most of the related prior work.

Reproducibility: No

Additional Feedback:

[Author Response · NeurIPS 2020]

We thank all reviewers for all the constructive comments. We are encouraged that they found our contribution novel and significant toward feature selection and explainable AI (R4). We are glad they found the methods rigorously presented with clear and formal definitions (R1), the theoretical grounding was clear and easy to follow, and also they found that the approach is general enough to be of wide interest to the NeurIPS community (R2). Moreover we are pleased that they found our result on gene expression important (R2,R3). We address reviewers minor comments below and will incorporate all feedback in our revised version.

---

[R1] **Complex architecture, non-convex optimization, lack of guarantee in solutions obtained in practice (local minima etc).** We maximize the variational lower bound of Mutual Information (MI) as a surrogate to maximizing MI itself. The approximate distributions are modeled using a flexible neural network and optimized via stochastic gradient descent (SGD). Therefore, while the objective is non-convex, our solution is still guaranteed to reach the local minimum. [R1] **Time Complexity**: Since most competing deep neural net methods also used SGD, our complexity are comparable to these deep models and scalable to sample size. We have also observed a comparable run-time experimentally, but choose to highlight the interpretability experiments. We'll add run-time in the appendix.

---

[R2] **Gene expression Fig. 5 and 6 results.** As described by the captions of Fig. 5 and 6, they are not figures of the gene expression, nor is it a correlation matrix. Fig. 5 displays an indicator matrix (white selected; black not) telling us which genes *work together* to best predict the smoking status for each patient. Based on Def. 1 and 2, we discovered the features that are most dependent on each other and the label. Note that Fig. 5 indeed shows that the genes in the first top rows are selected by non-smokers and the genes in the lower rows are selected by smokers. Note that this cannot be discovered by correlation feature clustering alone because it gives the same cluster of features (groups) for all samples. [R2] **Direct ref for the Gumbel Softmax trick?** Yes, we have cited Ref. [22] and provided a background review in the paper for completeness. [R2]**Other Methods to Compare.** Note that we have compared our method to *nine* competing methods from various perspectives to feature selection (FS): global FS, deep instance-wise FS, deep unsupervised FS, and global FS with feature group learning (includes feature group clustering). We can add results on attention models to Fig. 7 in the appendix. They provide focus areas in the image rather than all the similarly important features (pixels).

---

[R3] **Thms 1 and 2.** $I(\hat{\mathbf{X}}; \mathbf{X})$ **is upper bounded by the** $H(X)$. **Is the upper bound reachable?** Theorem 1 and 2 states that $I(\hat{\mathbf{X}}; \mathbf{X}) = I(\mathbf{X}; \mathbf{X}) \iff I(\mathbf{X}; \mathbf{X}|\mathbf{Z}) = 0$ and $I(\bar{\mathbf{X}}; \mathbf{Y}) = I(\mathbf{X}; \mathbf{Y}) \iff I(\mathbf{X}; \mathbf{Y}|\mathbf{Z} \odot \mathbf{s}) = 0$ respectively . This statement is not a guarantee of global optimum for LHS (i.e, $I(\hat{\mathbf{X}}; \mathbf{X})$), but rather a justification of why maximizing LHS can lead to desirable redundancies in RHS (i.e. $I(\mathbf{X}; \mathbf{X}|\mathbf{Z}) = 0$). By using neural network, we have experimentally shown (for genetic and MNIST) that it is sufficiently flexible to achieve highly accurate results. [R3]**Theorem 3 might not be useful.** We use Thm 3 to reveal how our definition of redundancy can be used to reveal the appropriate number of feature groups (a difficult issue in any clustering problem). It also supplies the theoretical background for us to discuss Thm 4, where the required assumption matches many genetic and public datasets. [R3] **Lasso Performing Better.** For the 2 cases in synthetic dataset where the correlation pattern was global, Lasso performed better; but when there is a mixture of correlation patterns their accuracy decreased to 74 percent. In general, such as in biology and image, data often have wide variations even within the same class. Our method is able to capture both patterns at a cost of a slight accuracy degradation (99.7% and 95% acc) on simple models. [R3] **Mutual Information (MI) Lower Bound.** We show in App. G that the objective $\max_{\theta_G, \theta_S} I(\mathbf{Y}; \bar{\mathbf{X}})$ can be alternatively maximized with $\max_{\theta_G, \theta_S} E_{\mathbf{Y}, \bar{\mathbf{X}}}[\log P(\mathbf{Y}|\bar{\mathbf{X}})]$ via Monte Carlo estimation, drawing samples from $P(\mathbf{Y}, \bar{\mathbf{X}}) = P(\mathbf{Y}|\bar{\mathbf{X}})P(\bar{\mathbf{X}})$. This requires us to obtain samples from $\bar{\mathbf{X}}$ and know the distribution $P(\mathbf{Y}|\bar{\mathbf{X}})$. Since $P(\mathbf{Y}|\bar{\mathbf{X}})$ is not known, we then approximate this distribution via a neural network $Q_{\theta_P}$. Eq. (8) is necessary because it provides the ancestral sampling steps to obtain samples for $\bar{\mathbf{X}}$. A similar logic then follows for Eq. (7) and the distribution $P(\mathbf{X}, \hat{\mathbf{X}})$. We will adjust this section to better explain the process.

---

[R3, R4]**Literature review on MI.** L2X, that we cite and learns instance-wise feature selection (FS), is based on MI. Many traditional (global) FS utilizes MI as criterion for selection(as it is a natural criterion for measuring dependency among random variables), including mRMR. mRMR maximizes feature relevance while minimizing feature redundancy to find the *global minimal subset of features*. In contrast, our method, learns *instance-wise group FS*. We differ in two ways: (1) In mRMR, if $F_1$ and $F_2$ are highly dependent, it'll only pick one of them if they are relevant to prediction. If $F_1$ and $F_2$ are relevant to prediction, we select both as a group (highlighting to domain scientists that these two features are both relevant and redundant to each other). (2) mRMR is global – all samples select the same features; our method is instance-wise – we provide which feature groups are important to each sample. Thank you for the advice; we'll include mRMR and MI-based FS in the revision. [R4] **How to stop each instance obtaining group of features and overwhelm users in terms of cognitive attention:** All the groups are being learned through a neural network which is a continuous mapping function, hence if two samples are very similar, the model would give a similar representation for the groups that are important for the prediction.

[Meta-Review · NeurIPS 2020]

In this paper, authors propose an instance-wise feature grouping, which is an interesting research topic. The proposed algorithm is backed up with theoretical analysis, and authors compared the proposed algorithm with several existing methods. However, there exist some concerns about experiments and did not fully addressed. I strongly encourage authors to address the following issues in the final version. 1. Validating the approach on gene expression (comments from Reviewer#2) The feature grouping per instance provided something novel for gene expression. (or some other dataset) 2. Comparison to attention-based learning methods. Therefore, I recommend weak acceptance for this paper. If this paper is not accepted, it is simply due to the lack of slots for this time.